# Scalable synthesis of ant-nest-like bulk porous silicon for high-performance lithium-ion battery anodes

Weili An [1,2], Biao Gao[1,3], Shixiong Mei[1], Ben Xiang[1], Jijiang Fu[1], Lei Wang[2], Qiaobao Zhang[4], Paul K. Chu [3] & Kaifu Huo [2]

Although silicon is a promising anode material for lithium-ion batteries, scalable synthesis of silicon anodes with good cyclability and low electrode swelling remains a significant challenge. Herein, we report a scalable top-down technique to produce ant-nest-like porous silicon from magnesium-silicon alloy. The ant-nest-like porous silicon comprising three-dimensional interconnected silicon nanoligaments and bicontinuous nanopores can prevent pulverization and accommodate volume expansion during cycling resulting in negligible particle-level outward expansion. The carbon-coated porous silicon anode delivers a high capacity of 1,271 mAh $g^{-1}$ at 2,100 mA $g^{-1}$ with 90% capacity retention after 1,000 cycles and has a low electrode swelling of 17.8% at a high areal capacity of 5.1 mAh $cm^{-2}$. The full cell with the prelithiated silicon anode and $Li(Ni_{1/3}Co_{1/3}Mn_{1/3})O_2$ cathode boasts a high energy density of 502 Wh $Kg^{-1}$ and 84% capacity retention after 400 cycles. This work provides insights into the rational design of alloy anodes for high-energy batteries.

[1] The State Key Laboratory of Refractories and Metallurgy and Institute of Advanced Materials and Nanotechnology, Wuhan University of Science and Technology, 430081 Wuhan, China. [2] Wuhan National Laboratory for Optoelectronics (WNLO), Huazhong University of Science and Technology, 430074 Wuhan, China. [3] Department of Physics and Department of Materials Science and Engineering, City University of Hong Kong, Tat Chee Avenue, Kowloon 999077 Hong Kong, China. [4] Department of Materials Science and Engineering, College of Materials, Xiamen University, 361005 Xiamen, Fujian, China. These authors contribute equally: Weili An, Biao Gao. Correspondence and requests for materials should be addressed to Q.Z. (email: zhangqiaobao@xmu.edu.cn) or to K.H. (email: kfhuo@hust.edu.cn)

Fast development of portable electronic devices and electric vehicles requires lithium-ion batteries (LIBs) with higher specific power/energy, longer cycle life, and competitive costs[1,2]. Silicon (Si) has been identified as one of the promising anode materials for next-generation high-energy density LIBs because of its large theoretical specific capacity of 3579 mAh g$^{-1}$ ($Li_{15}Si_4$)[3–5]. However, Si suffers from a large volume change (>300%) during lithiation and delithiation causing mechanical pulverization of the particles, loss of inter-particles electrical contact, and continuous formation of the solid-electrolyte interface (SEI), consequently resulting in rapid capacity fading and deteriorated battery performance[5–11].

Progress has been made to address particle pulverization by decreasing the size to the critical nanosize[10]. Si nanostructures such as nanoparticles[12], nanowires[13], nanotubes[9], as well as nano-Si/carbon hybrids[6,14] have been developed as anode materials and enhanced cycle life compared to the bulk counterparts have been demonstrated. However, scalable synthesis of nanostructured Si with a large tap density, high initial Coulombic efficiency (ICE), and long cycle stability at a high mass loading remains a challenge[5,10]. Nanostructured Si has a large surface area, which increases the electrode/electrolyte interfacial area giving rise to low ICE and the small tap density causes a low volumetric energy density. To improve the tap density and ICE without structural pulverization during lithiation/delithiation, microscale or porous Si particles assembled from nanoscale building blocks have been proposed as anodes in LIBs[11,15–20]. For example, Cui et al[20]. prepared yolk-like nanoscale Si/C assembled microscale pomegranate-like particles, which had a tap density of 0.53 g cm$^{-3}$ and 97% capacity retention after 1000 cycles. Park et al[16]. prepared microscale porous Si by electroless metal deposition and chemical etching and the materials had a high capacity of 2050 mAh g$^{-1}$ at 400 mA g$^{-1}$. However, fabrication of these microscale Si or Si/C materials tends to be costly and is not yet scalable due to the complex synthesis. The tap density and electrochemical cyclability are still unsatisfactory from the commercial standpoint[21]. More importantly, large thickness swelling of the Si anodes presents the most critical challenge hampering practical implementation in high-energy full cells[22] but this issue is often ignored in Si anode research. The large electrode swelling in thickness during lithiation in LIBs decreases the volumetric energy density and undermines cycling performance and unsafety[23]. To address these limitations, rational design of Si anode materials with a large tap density, minimal electrode thickness swelling, and large areal capacity (>3.0 mAh cm$^{-2}$ for commercial LIBs[24]) together with a cost-effective and scalable preparation method are highly desirable but still very challenging.

Herein, we report an ant-nest-like microscale porous Si (AMPSi) for high-performance anodes in LIBs. The AMPSi is produced via a low-cost and scalable top-down approach by thermal nitridation of the Mg-Si alloy in nitrogen ($N_2$) followed by the removal of the $Mg_3N_2$ by-product in an acidic solution (schematically shown in Fig. 1a). The synchrotron radiation tomographic reconstruction images reveal that the AMPSi has 3D interconnected Si nanoligaments and bicontinuous nanoporous network resembling the natural ant nest (Fig. 1b). In situ transmission electron microscopy (TEM) reveals that the Si nanoligaments with widths of several 10 nm can expand/shrink reversibly during lithiation/delithiation without pulverization and the volume expansion of the Si nanoligaments can be accommodated by the surrounding pores through reversible inward Li breathing, thereby resulting in negligible particle-level outward expansion as schematically shown in Fig. 1c. The AMPSi integrates the intrinsic merits of nanoscale and microscale Si with a high tap density of 0.84 g cm$^{-3}$ and small surface area. After coating a 5–8 nm thick carbon layer to improve the conductivity, the anode composed of carbon-coated AMPSi (AMPSi@C) shows a high ICE of 80.3%, gravimetric capacities of 2134, and 1271 mAh g$^{-1}$ at 0.1 and 0.5 C rates (1 C = 4200 mA g$^{-1}$), and 90% capacity retention from the twentieth to thousandth cycles. The AMPSi@C anode with an areal mass loading of 0.8 mg cm$^{-2}$ delivers a large volumetric capacity of 1712 mAh cm$^{-3}$ at 0.1 C after 100 cycles, which is the highest reported from Si anodes so far. Furthermore, the bulk electrode of AMPSi@C with an areal capacity of 5.1 mAh cm$^{-2}$ exhibits a small electrode swelling of 17.8%, which substantially outperforms most reported Si anodes[11,15,25–27]. The full cell comprising the prelithiated AMPSi@C anode and commercial $Li(Ni_{1/3}Co_{1/3}Mn_{1/3})O_2$ cathode has a high-energy density of 502 Wh Kg$^{-1}$ and long-life cycle stability with 84% capacity retention for over 400 cycles. The economic and scalable top-down fabrication method, rational bulk nanoporous structure design, as well as superior electrochemical properties can be extended to other types of electrodes that tend to undergo large volume expansion in high-energy batteries.

## Results

**Synthesis and characterization of AMPSi and AMPSi@C.** As shown in Fig. 1a, fabrication of AMPSi commences with $Mg_2Si$

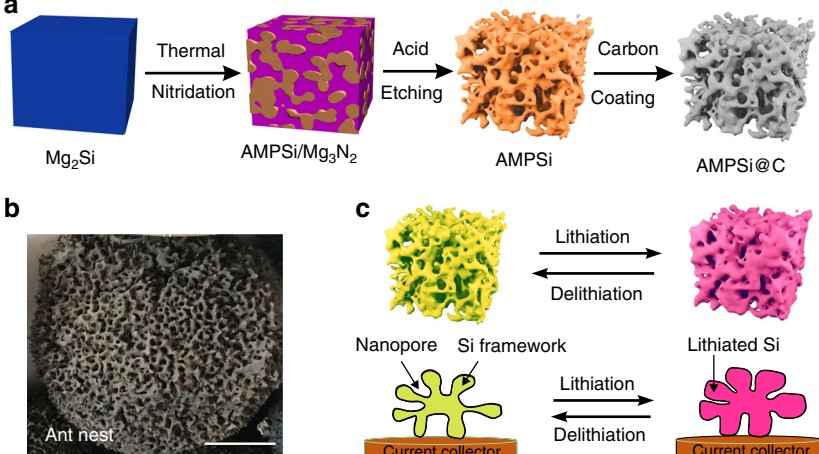

**Fig. 1** Design and schematic showing the synthesis method of AMPSi. **a** Schematic showing the preparation of AMPSi and AMPSi@C. **b** Photograph of an ant nest (scale bar = 20 cm). **c** Schematic illustrating the lithiation/delithiation process of the ant-nest-like microscale porous Si particles showing inward volume expansion and stable Si framework retention during cycling

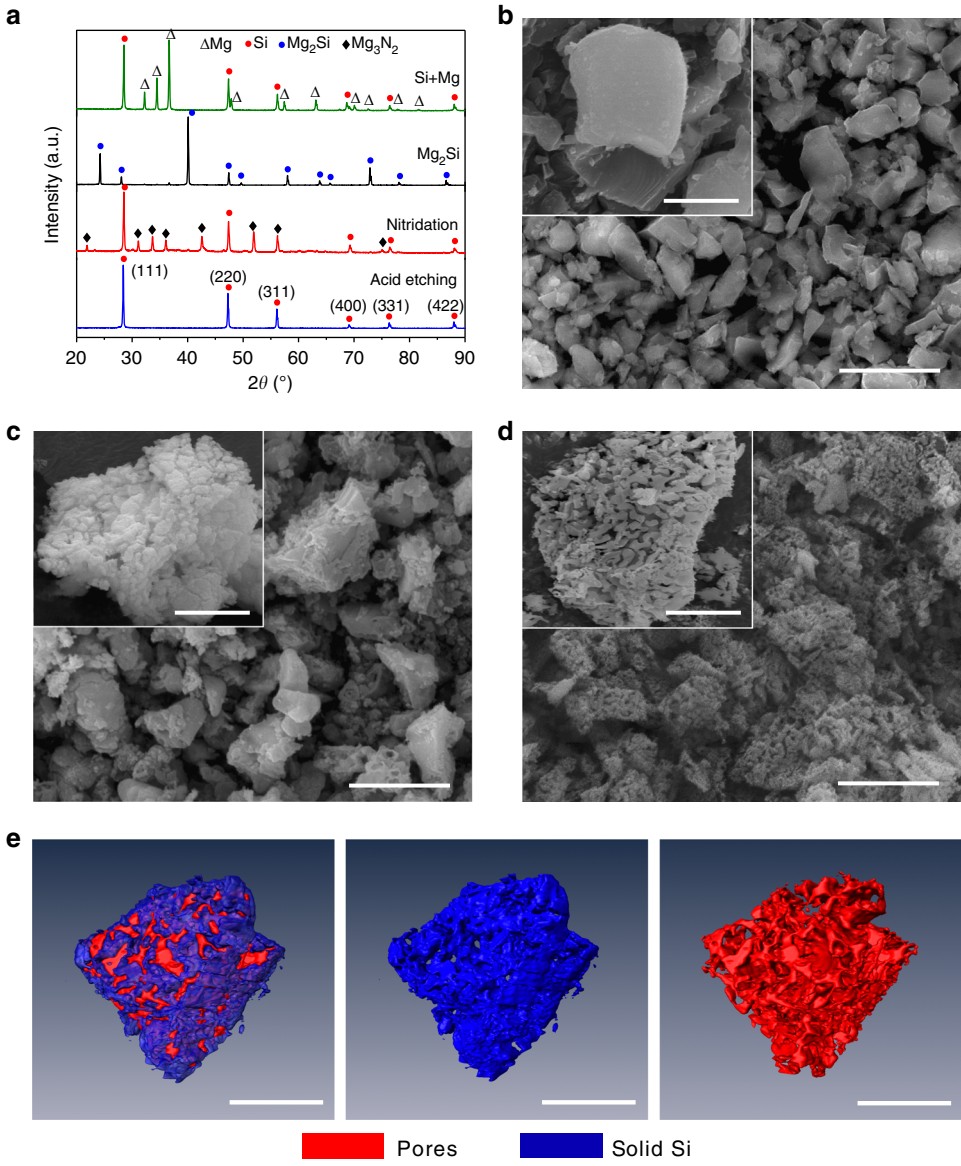

**Fig. 2** Morphological and structural characterization. **a** XRD patterns of the products at different steps during preparation. SEM images of **b** pristine Mg$_2$Si particles, **c** nitrided Mg$_2$Si particles, and **d** AMPSi (scale bar for **b**, **c**, and **d** = 3 μm and scale bar for the insets = 2 μm). **e** Synchrotron radiation tomographic 3D reconstruction images of the AMPSi (scale bar = 3 μm)

powders commercially available or synthesis from Mg and metallurgical Si. Here the crystalline 3–5 μm Mg$_2$Si particles are prepared by alloying the bulk metallurgical Si and Mg at 550 °C and then the as-obtained Mg$_2$Si particles are nitrided under N$_2$ at 750 °C. During this process, Mg in Mg$_2$Si reacts with N$_2$ to produce Mg$_3$N$_2$ while Si is separated forming the Mg$_3$N$_2$/Si composite $(3Mg_2Si\ (s) + 2N_2\ (g) \rightarrow 3Si\ (s) + 2Mg_3N_2\ (s))$. The X-ray diffraction (XRD) patterns in Fig. 2a exhibit peaks from Mg$_3$N$_2$ (JCPDS No. 73–1070) and Si (JCPDS No. 27–1402), while the peaks associated with the Mg$_2$Si phase (JCPDS No. 35–0773) disappear. Compared to pristine Mg$_2$Si particles (Fig. 2b), the nitrided Mg$_2$Si particles become coarse (Fig. 2c) and the high-angle annular dark-field (HAADF) scanning transmission electron microscopy (STEM) and TEM images reveal a loose connecting framework (Supplementary Fig. 1a, b). High-resolution TEM (HR-TEM) (Supplementary Fig. 1c) discloses that the single-crystal Mg$_3$N$_2$ and Si are closely connected forming the Mg$_3$N$_2$/Si heterostructure. Energy-dispersive X-ray spectroscopy (EDS) mapping (Supplementary Fig. 1d-g) shows uniform

distributions of Si, Mg, and N. After removing Mg$_3$N$_2$ in diluted hydrochloric acid, the bulk porous Si particles (Fig. 2d) are obtained. The size of the synthesized AMPSi is measured by a laser particle size analyzer (Mastersizer 2000) and the average diameter ($D_{50}$) is 3 ± 0.2 μm (Supplementary Fig. 2a). No Mg signal is detected from AMPSi by X-ray photoelectron spectroscopy (XPS) indicating complete removal of Mg (Supplementary Fig. 3a, b). The Si 2p XPS spectra in Supplementary Fig. 3c, d show two peaks corresponding to Si 2p 1/2 and Si 2p 3/2 of elemental Si (Si$^0$) at binding energies of 98.8 and 99.4 eV, and the weak ones at 102.2 and 102.8 eV are associated with SiO$_x$ formed by native oxidation after synthesis[13–15]. The oxygen content in AMPSi is about 6.7% (wt.). The magnified scanning electron microscopy (SEM) image in the inset of Fig. 2d shows that the porous Si particle consists of interconnected nanoligaments and 3D bicontinuous nanopores resembling the natural ant nest (Fig. 1b). The microstructure of AMPSi is further analyzed by TEM as shown in Supplementary Fig 2b, c. The Si ligaments with a size of 30–50 nm are interconnected and surrounded by the

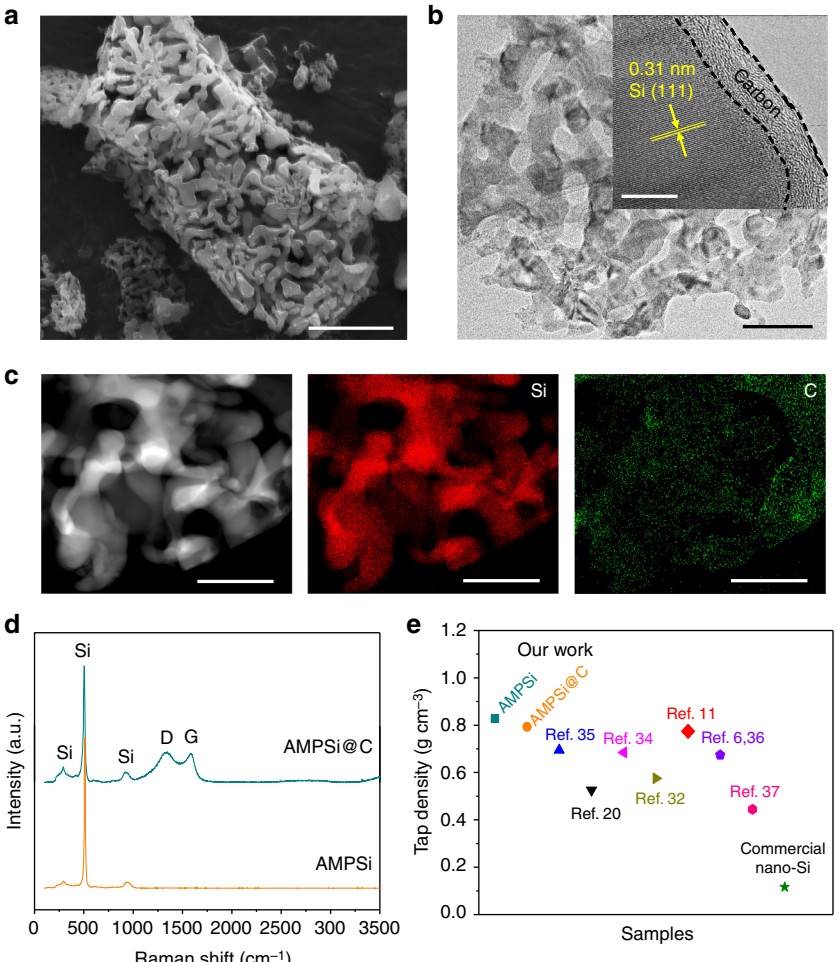

**Fig. 3** Characterization of AMPSi@C. **a** SEM image of AMPSi@C (scale bar = 2 μm). **b** TEM image of AMPSi@C (scale bar = 100 nm). The inset (scale bar = 10 nm) is the HR-TEM image showing that the 5–8 nm thickness amorphous C shell is coated on the Si nanoligaments and the lattice distance of 0.31 nm corresponds to the $d$-spacing of the (111) planes of crystalline Si (111). **c** EDS maps of the Si frameworks in AMPSi@C with red and green corresponding to Si and C, respectively (scale bar for **c** = 200 nm). **d** Raman scattering spectra of AMPSi and AMPSi@C. **e** Comparison of the tap densities between our Si anodes and other Si-based anode materials (see Supplementary Table 1)

bicontinuous nanopores forming an ant-nest-like framework. The synchrotron radiation tomographic images (Fig. 2e) further confirm the 3D continuous nanopore structure and interconnected Si nanoligaments. This unique ant-nest-like material are fundamentally different from previously reported Si/C composite secondary particles[11,15–20]. HR-TEM and selected-area electron diffraction (SAED) reveal that AMPSi is composed of many crystalline grains of Si (Supplementary Fig. 4). The AMPSi has a high tap density of 0.84 g cm$^{-3}$, which is larger than that of previously reported nanostructured Si and microscale Si[6,11,14,15,20,25–27].

Since the well-organized pores are continuous and most of the pores are larger than 50 nm, mercury intrusion porosimetry is employed to examine the pores larger than 50 nm and nitrogen adsorption-desorption isotherms are obtained to assess the mesopores. Brunauer-Emmett-Teller (BET) analysis reveals that the AMPSi has a small specific surface area of 12.6 m$^2$ g$^{-1}$ due to the space-efficient packing of AMPSi (Supplementary Fig. 5a, b). The porosity of AMPSi is measured to be 64.3% that is close to the theoretical value of 68.6% assuming that the sample undergoes no macroscopic volume change during dealloying Mg in Mg$_2$Si[17]. As schematically shown in Fig. 1c, the high porosity of AMPSi allows inward expansion of the 3D interconnected Si nanoligaments during cycling consequently,

leading to high structural stability and negligible particle-level outward expansion.

To improve the electrical conductivity, a thin C layer is coated on AMPSi by dopamine self-polymerization followed by thermal carbonization, as schematically shown in Fig. 1a. Dopamine is a widely used carbon precursor for C coatings. It can self-polymerize into polydopamine (PDA) coatings on the surface of Si with strong adhesion and so the thin and homogeneous carbon shell on Si can be achieved after thermal carbonization. The SEM and TEM images (Fig. 3a, b) disclose that the AMPSi@C retains the 3D ant-nest-like porous structure of the pristine AMPSi and Supplementary Fig. 5b shows that the pore size decreases slightly after C coating. To confirm the uniform carbon coating, HR-TEM is performed on AMPSi@C at different regions. The HR-TEM images at different regions (inset in Fig. 3b and Supplementary Fig. 6a, b) suggest that the C shell having a thickness of 5–8 nm is amorphous and the lattice distance of 0.31 nm corresponds to the $d$-spacing of the (111) planes of crystalline Si[25,28]. EDS mapping (Fig. 3c and Supplementary Fig. 6c) further confirms that C is uniformly coated on the surface of AMPSi. The Fourier transform infrared (FTIR) spectroscopy spectra of AMPSi and AMPSi@C are depicted in Supplementary Fig. 7. The bands at 1060 and 1620 cm$^{-1}$ correspond to the characteristic vibrations of Si. Compared to AMPSi, the band of AMPSi@C at 1060 cm$^{-1}$

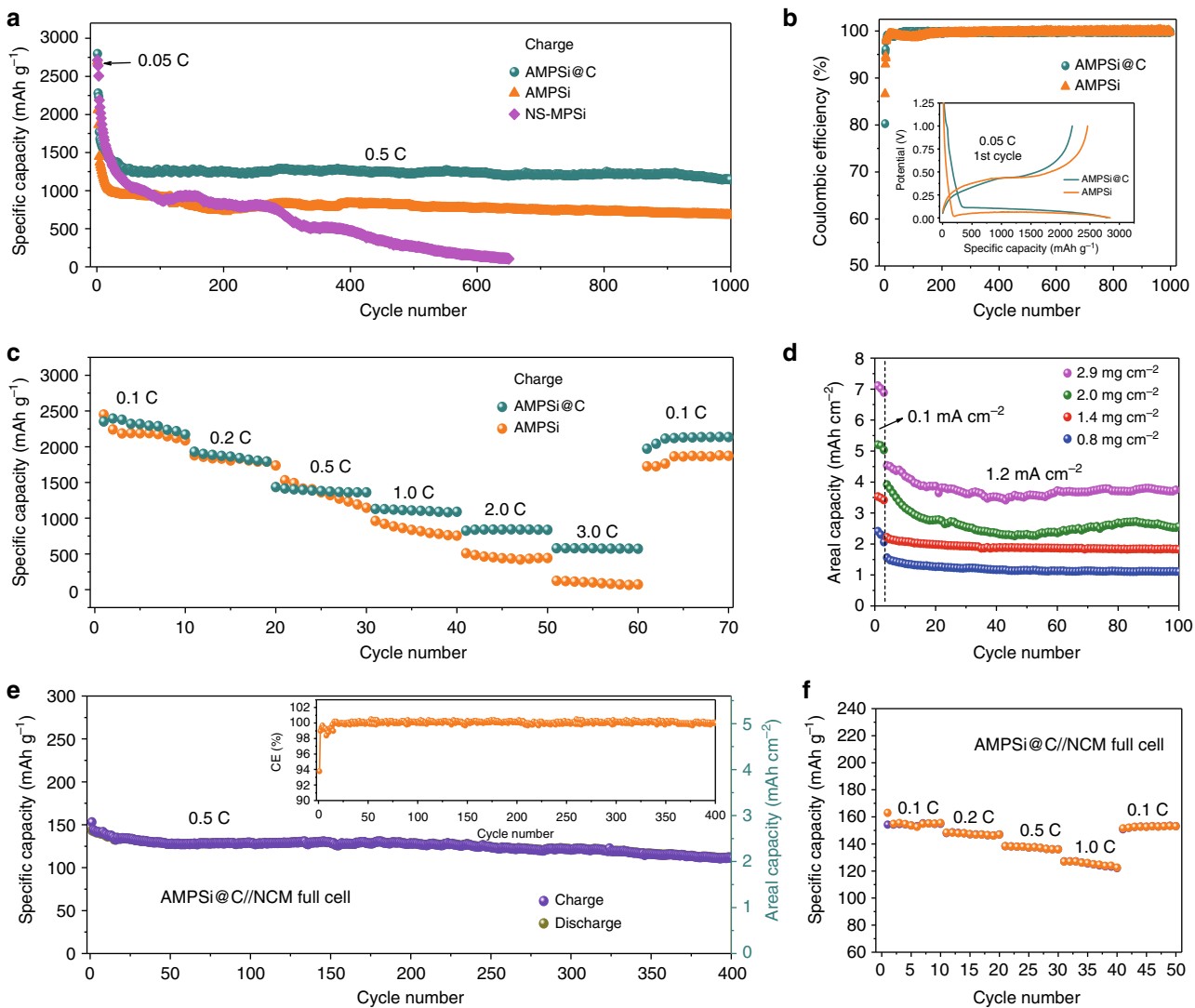

**Fig. 4** Electrochemical characterization of anodes in half-cell or full-cell configurations. **a** Long cycling test of AMPSi@C at 0.5 C after the activation process in the first three circles at 0.05 C (1 C = 4.2 A g$^{-1}$) in the half-cell with areal mass loading of 0.8 mg cm$^{-2}$. **b** Comparison of the CE between AMPSi and AMPSi@C in the half-cell configuration with the inset being the corresponding voltage profiles. **c** Rate performance of AMPSi@C and AMPSi at various current densities from 0.1 to 3 C. **d** Areal capacities vs. cycling number of the AMPSi@C anodes with different mass loadings at a current density of 1.2 mA cm$^{-2}$ (the initial three cycles are carried out at 0.1 mA cm$^{-2}$). **e** Full-cell charge at 0.5 C (1 C = 160 mA g$^{-1}$) with prelithiated AMPSi@C anode and a Li (Ni$_{1/3}$Co$_{1/3}$Mn$_{1/3}$)O$_2$ cathode. The inset showing the corresponding CE. **f** Rate performance of the full cell. All the specific capacity values in the half-cell are based on the total mass of AMPSi and C shell, unless otherwise stated

is split into two peaks at 1240 and 1090 cm$^{-1}$ suggesting robust bonding between Si and coated carbon shell in AMPSi@C[18].

The Raman scattering spectrum (Fig. 3d) of AMPSi shows a sharp peak at 511 cm$^{-1}$ and two weak peaks at 299 and 925 cm$^{-1}$, corresponding to the characteristic peaks of crystalline nanoscale Si[18,29,30]. The three peaks corresponding to Si are still clearly visible from AMPSi@C. However, the strong peak blue-shifts to 504 cm$^{-1}$ possibly due to confinement effect caused by the carbon coating and strong bonding between the coated carbon and Si[4]. The two peaks at 1347 and 1581 cm$^{-1}$ are attributed to the vibration modes of disordered graphite (D band) and E$_{2g}$ of crystalline graphite (G band) and the large I$_D$/I$_G$ ratio (1.14) reflects the low graphitic degree in the carbon coating consistent with XRD (Supplementary Fig. 8) and HR-TEM results. The thermogravimetric analysis shows that the C content in AMPSi@C is 8.5 wt% (Supplementary Fig. 9). The strong bonding between carbon and Si improves the cycle stability and stabilizes the SEI to enhance the CE during the charging/discharging cycles.

The tap density of AMPSi@C is measured to be 0.80 g cm$^{-3}$ and is slightly less than that of pristine AMPSi of 0.84 g cm$^{-3}$ due to the presence of amorphous C coating. However, it is still bigger than those of commercial nano-Si, microscale Si/C, and porous Si/C (Fig. 3e and Supplementary Table 1)[6,11,20,31–37] thus enabling a bigger volumetric energy density and thinner electrode thickness for the same mass loading in the practical cells.

**Electrochemical performance of AMPSi and AMPSi@C.** The electrochemical performance of the AMPSi and AMPSi@C electrodes is evaluated using half-cell and full-cell configurations, respectively. For comparison, nanoparticles assembled with 3D mesoporous Si (NS-MPSi) are also prepared by thermal distillation of Mg in Mg$_2$Si in vacuum[38]. The as-prepared NS-MPSi particles are composed of 20–40 nm primary particles with a specific surface area of 120 m$^2$ g$^{-1}$ (Supplementary Fig. 10). The electrolyte is 1.0 M LiPF$_6$ in 1:1 v/v ethylene carbonate/diethyl

carbonate with 6 vol % vinylene carbonate (VC) as the additive. VC is widely used as an electrolyte additive as it can boost the formation of a smooth and stable SEI on Si-based anodes[39]. All the specific capacity values shown in this paper are based on the total mass of AMPSi@C, unless otherwise stated.

The delithiation capacities of AMPSi, AMPSi@C, and NS-MPSi versus cycle number are presented in Fig. 4a. After the three-cycle activation step at C/20, the capacity of the AMPSi electrode is maintained at above 679 mAh g$^{-1}$ at 0.5 C with good cycle stability for over 1000 cycles. Under similar conditions, the NS-MPSi electrode shows rapid capacity decay from 2712 mAh g$^{-1}$ in the initial cycle to below 100 mAh g$^{-1}$ after 650 cycles mainly due to the electrode cracking and structural destruction of NS-MPSi during cycling (Supplementary Fig. 11). Compared to AMPSi, AMPSi@C shows a larger reversible capacity of 1271 mAh g$^{-1}$ with 90% capacity retention from the twentieth to thousandth cycles (Fig. 4a). The large capacity decay during the first 20 cycles is attributed to the increased current density from 0.05 to 0.5 C and continuous formation of SEI due to the slow penetration of the viscous organic electrolyte into the continuous porous structure in AMPSi@C as a result of strong capillary effects, volume expansion of Si, and low crystalline carbon coating. Actually, capacity decay during the first several and even tens of cycles have been observed from Si and other anode materials[9,20,32,33]. The dQ/dV curve of AMPSi@C is presented in Supplementary Fig. 12 and the anodic and cathodic peaks overlap after 20 cycles suggesting high cycle reversibility. At a high rate up to 3 C, AMPSi@C electrode still shows a large reversible capacity of 632 mAh g$^{-1}$ with 0.015% capacity decay per cycle for over 1000 cycles (Supplementary Fig. 13). Moreover, 72.5% capacity retention can be achieved after 500 cycles at a higher rate of 5.0 C (Supplementary Fig. 14), which outperforms most previously reported microscale Si and amorphous-C-coated Si anodes (Supplementary Tables 1 and 2). The role of Fluoroethylene carbonate (FEC) and VC electrolyte additives is also investigated. Although the AMPSi@C anodes display similar cycle stability for both electrolyte additives, AMPSi@C has a slight larger average CE with VC additive (Supplementary Fig. 15).

Cyclic voltammetry (CV) is performed on AMPSi@C in a potential range of 0.01–1.0 V versus Li/Li$^+$ at a scanning rate of 0.1 mV s$^{-1}$ as shown in Supplementary Fig. 16a. The broad cathodic peak at ~0.16 V is ascribed to the formation of Li$_x$Si and the anodic peaks at 0.40 and 0.53 V are characteristic of the Li desertion process from Li$_x$Si to amorphous Si. The chemical states of the cycled electrode are determined by XPS. The fine XPS Si 2p results of the lithiated (0.01 V) and delithiated (0.40 and 0.53 V) samples during the first cycle are depicted in Supplementary Fig. 17. The pristine AMPSi@C shows two Si peaks at 98.8 and 99.4 eV assigned to Si 2p1/2 and Si 2p3/2 of elemental Si (Si$^0$). After full lithiation to 0.01 V, the two Si peaks shift to 97.3 and 97.9 eV due to the alloying reaction to form the Li$_x$Si phase. When the electrode is delithiated at 0.40 V, the peaks of Si$^0$ reappear and those corresponding to Li$_x$Si shift to high binding energy, suggesting partial Li-ion desertion from the Li$_x$Si alloy. At a larger delithiation voltage of 0.53 V, stronger peaks of Si$^0$ are observed in line with the Si binding energy in Li$_x$Si shifting to higher energy, meaning that the decreased Li content in Li$_x$Si stems from more Li-ion desertion from the Li$_x$Si alloy. The ex situ XPS results agree well with the CV data confirming the alloying/dealloying reactions of AMPSi@C during lithiation/delithiation. The peak intensity in the CV curves of AMPSi and AMPSi@C (Supplementary Fig. 16a, b) increases initially with cycling possibly due to gradual activation of the electrodes[20]. After several cycles, these peaks overlap suggesting high reversibility and stability. The AMPSi@C electrode exhibits higher peak

currents in comparison with the AMPSi electrode implying a largely enhanced capacity. Moreover, the anodic peaks of the AMPSi@C electrode shift to higher voltage while the cathodic peaks shift to a lower voltage. The improved capacity and smaller voltage separation (Supplementary Fig. 16a, b) stem mainly from the higher Si electrochemical utilization ratio and enhanced electrical conductivity of AMPSi@C, which is further confirmed by electrochemical impede spectroscopy as shown in Supplementary Fig. 18a. The Nyquist plots of the AMPSi and AMPSi@C and corresponding equivalent circuit are depicted in Supplementary Fig. 18. The two semicircles in the high-frequency region represent the resistance of the SEI film (R$_{sf}$) and charge transfer resistance (R$_{ct}$) and the straight lines in the low-frequency region correspond to diffusion of lithium ions (Z$_w$). R$_{sf}$ of AMPSi@C is smaller than that of AMPSi, indicating that the SEI at AMPSi@C is thinner and more stable than that in AMPSi. Moreover, R$_{ct}$ of AMPSi@C is less than that of AMPSi suggesting smaller resistance due to the high conductivity of the carbon coating. The voltage profiles of the AMPSi and AMPSi@C electrodes during initial cycling are shown in the inset in Fig. 4b, which shows the typical lithiation plateau at around 0.1 V corresponding to the alloying reaction of Si with Li to form Li$_x$Si alloy. The initial lithiation capacity of both the AMPSi and AMPSi@C electrodes reaches 2843 mAh g$^{-1}$ at 0.05 C indicating that most of the Si in AMPSi and AMPSi@C is active due to the high Li$^+$ accessibility of the bulk nanoporous structure. The ICE of AMPSi is 86.6% (Fig. 4b) and CE reaches 99.9% after 10 cycles, which are better than those of Si/C composites[4,10,12]. The ICE of AMPSi@C is slightly reduced (80.3%) possibly because of the formation of more SEI on the surface of the amorphous C shell as a result of the enlarged surface area of AMPSi@C compared to AMPSi[18]. Figure 4c presents the rate performance of the AMPSi and AMPSi@C electrodes. Although both electrodes have similar capacities at 0.1 and 0.2 C, AMPSi@C has much higher capacities at higher rates. Even at a high rate of 3 C, the AMPSi@C electrode has a high capacity of 619 mAh g$^{-1}$ that is almost twice that of the AMPSi electrode. Moreover, when the current density is reverted to 0.1 C, a reversible capacity of 2134 mAh g$^{-1}$ is recovered readily implying high structural stability of the AMPSi@C electrode. The tap density of the AMPSi@C electrode is 0.80 mg cm$^{-3}$ and the volumetric capacity of the AMPSi@C with an areal mass loading of 0.8 mg cm$^{-2}$ after 100 cycles is measured to be 1712 mAh cm$^{-3}$ (the lithiated stage) at 0.1 C rate, which is the best value reported from Si-based electrodes so far (Supplementary Fig. 19a)[21,40–44].

The areal capacities of the AMPSi@C electrodes with different areal mass loadings from 0.8 to 2.9 mg cm$^{-2}$ are shown in Fig. 4d. The capacity of the AMPSi@C anode increases linearly with areal mass loading (Supplementary Fig. 19b), indicating that the Si active materials are utilized effectively in spite of the large mass loading. At a high areal mass loading of 2.9 mg cm$^{-2}$, the areal capacities of AMPSi@C reach 7.1 mAh cm$^{-2}$ at 0.1 mA cm$^{-2}$ and 3.9 mAh cm$^{-2}$ at 1.2 mA cm$^{-2}$ after 100 cycles. These values are higher than those of most of the reported Si anodes[5,10,14,26,45–48]. Generally, when the mass loading of Si is increased to achieve a larger areal capacity, cycle performance tends to worsen due to the increased serial resistance of the particle-electrolyte interface and electrode-level disintegration[49]. However, the AMPSi@C electrode with a high mass loading still displays good stable cycle stability and excellent rate characteristics (Supplementary Fig. 19c, d) confirming that bulk nanoporous structure of AMPSi@C electrode is favorable to Li$^+$ accessibility and electron transport.

To further evaluate the practicality of AMPSi@C in LIBs, a full battery is assembled with the commercial Li(Ni$_{1/3}$Co$_{1/3}$Mn$_{1/3}$)O$_2$ (NCM) cathode and prelithiated AMPSi@C anode. The full-cell test is done with an anode limited capacity ratio of 1.1:1

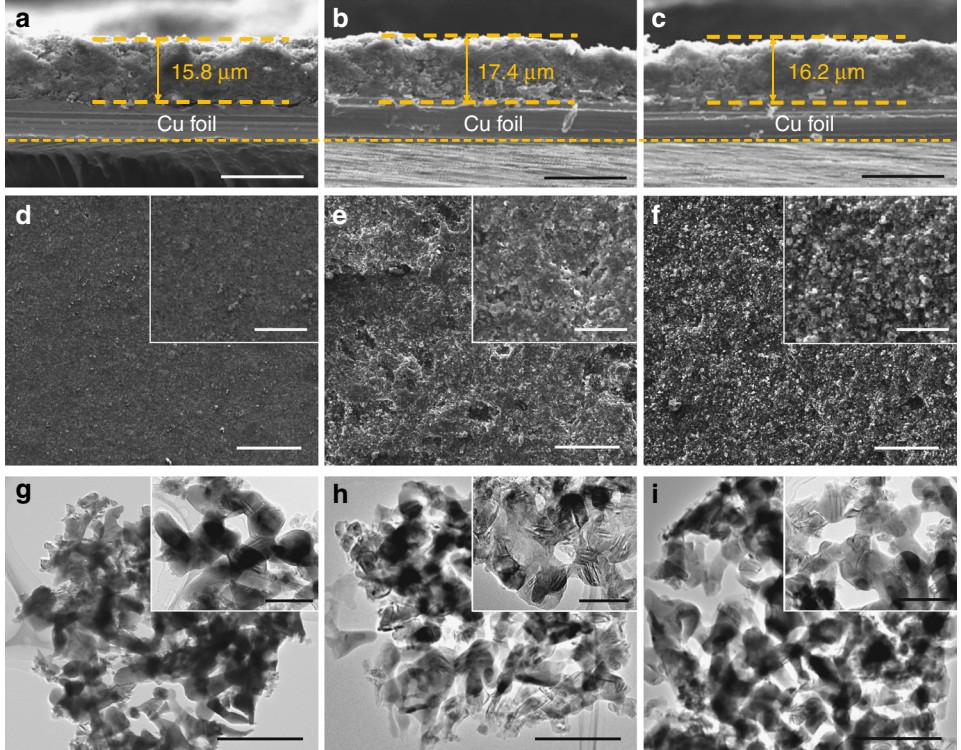

**Fig. 5** Electrode swelling measurements of AMPSi@C. Cross-sectional SEM images of the AMPSi@C electrode films **a** before cycling, **b** after full lithiation, and **c** delithiation, respectively (scale bar for **a**, **b**, and **c** = 20 μm). **d**–**f** Corresponding top-view SEM images (scale bar for **d**– f = 100 μm and scale bar for insets = 20 μm). **g**–**i** TEM images of AMPSi@C, lithated AMPSi@C and delithiated AMPSi@C (scale bar for **g**– i = 1 μm and scale bar for insets = 100 nm)

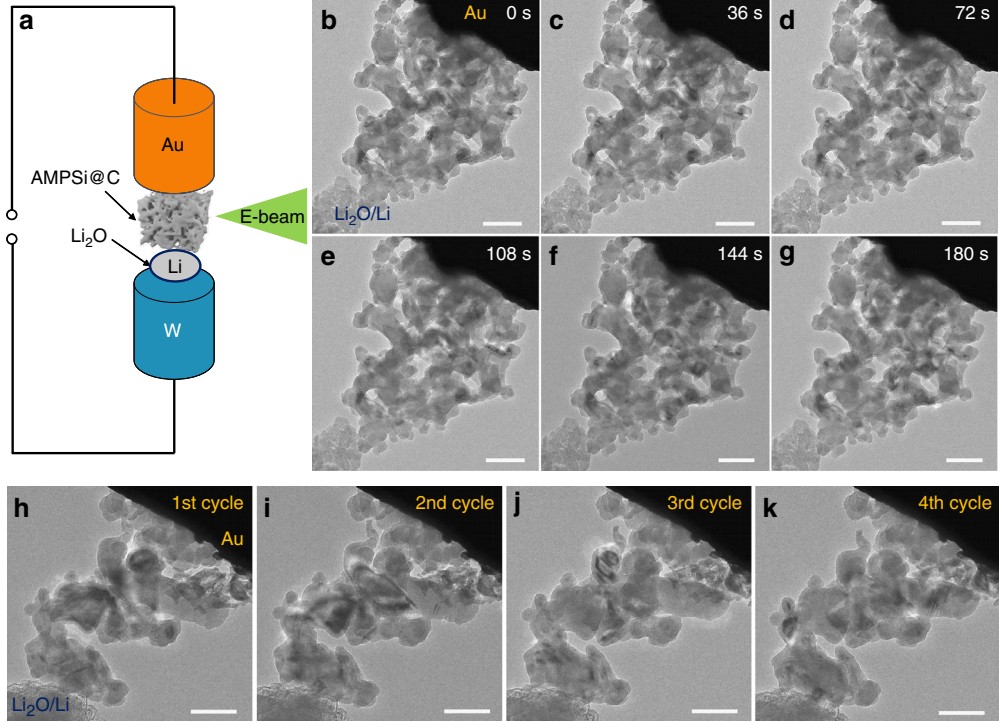

**Fig. 6** In situ lithiation/delithiation behavior of AMPSi@C. **a** Schematic of in situ nanobattery configuration. **b**–**g** Time-resolved TEM images depicting the lithiation process of AMPSi@C electrode (Supplementary Movie 1). After full lithiation, AMPSi@C exhibits inward expansion and does not show cracks and pulverization. **h**–**k** In situ TEM images of AMPSi@C after different cycles (Supplementary Movie 2) showing a stable structure during repeated cycling. Scale bar for **b**–**k** = 200 nm

considering safety and capacity matching of the full cell during cycling[50]. The prelithiation procedure is conducted in a half-cell by the first discharging process and the working electrode (AMPSi@C) is lithiated to 0.01 V at a 0.05 C rate by the galvanostatic discharging method (see details in Methods). The typical charging-discharging curves of the AMPSi@C//NCM full cell with a cutoff voltage of 2.80–4.25 V is shown in Supplementary Fig. 20a and Fig. 4e depicts the cycle stability. The AMPSi@C//NCM full cell delivers a high reversible capacity of 134 mAh g$^{-1}$ at 0.5 C (1 C = 160 mA g$^{-1}$ based on cathode active material) with capacity retention of 84% for over 400 cycles. Moreover, the full battery exhibits a high rate capability of 118 mAh g$^{-1}$ at 1.0 C (Fig. 4f). The corresponding CE is shown in the inset in Fig. 4e. The ICE of the AMPSi@C//NCM full cell is 94% and the later CE reaches 99.9% after 10 cycles (inset of Fig. 4e). The full cell has an average voltage of 3.75 V and the discharge capacity is 134 mAh g$^{-1}$ at 0.5 C, thus, the full cycle can deliver a high-energy density of 502 Wh kg$^{-1}$ outperforming previously reported Si-based full cells[4,46] (Supplementary Table 1 and 2). For comparison, we also evaluate the cycling performance of full cells using AMPSi@C anodes without prelithiation (Supplementary Fig. 20b). It shows a lower ICE of 83.1%, lower capacity, and poor cycle stability than the prelithiated AMPSi@C anode, indicating that prelithiation is necessary to enhance the performance of the full cell.

**Electrode swelling and in situ lithiation of AMPSi@C.** Electrode swelling is a show stopper for commercial implementation of Si-based LIBs but often ignored in Si anode research. Large electrode swelling of Si anodes undermines the long-term cycling stability and safety. The cross-sectional SEM images show that the pristine AMPSi@C electrode film has an average thickness of 15.8 μm (Fig. 5a) and after full lithiation, the electrode thickness increases to 17.4 μm (Fig. 5b) with 10.1% thickness swelling. After full delithiation, the electrode thickness recovers to 16.2 μm (Fig.5c), indicating negligible thickness change. The small swelling and negligible thickness fluctuation of the AMPSi@C electrode ensure superior stability and safety in practice. The corresponding top-view SEM images reveal that the AMPSi@C electrode shows no observable mechanical damage such as cracking or fracture during cycling (Fig. 5d–f). More importantly, the microstructure of AMPSi@C remains intact even after 1000 cycles, confirming the high structure stability and integrality during cycling as shown by the TEM images (Fig. 5g–i). Electrode thickness swelling of the Si anodes with different areal capacities of 3.5–7.1 mAh cm$^{-2}$ is also assessed (Supplementary Fig. 21). The AMPSi@C electrodes with thicknesses of 25.8, 34.7, and 45.1 μm deliver areal capacities of 3.5, 5.1, and 7.1 mAh cm$^{-2}$. After full lithiation, the electrode thicknesses increase to 29.4, 40.9, and 55.3 μm, corresponding to 14.0%, 17.8%, and 22.6% electrode swelling, respectively, as shown in Supplementary Fig. 22. These values are much smaller than those of previously reported Si anodes[11,15,25–27,47,48] (Supplementary Table 1). The large areal capacity and small electrode thickness swelling suggest promising applications of AMPSi@C in high-energy LIBs.

The structural evolution of AMPSi@C during lithiation is observed by in situ TEM using a nanobattery setup as schematically shown in Fig. 6a. AMPSi@C is attached on a gold tip and then connected to Li/Li$_2$O on a tungsten (W) tip. The as-formed thin Li$_2$O layer on Li serves as a solid electrolyte. The AMPSi@C is lithiated when a negative bias (−3 V) is applied to the W end and the delithiation process of AMPSi@C is realized when applying a positive bias. When the Li source comes in close contact, Li ions diffuse quickly from the contact point to the AMPSi@C via wave propagation-like motion[51]. The structural

changes of AMPSi@C during lithiation is monitored by the time-resolved TEM images in Fig. 6b–g captured from in situ videos (Supplementary Movie 1). As lithiation proceeds, the Si framework maintains its intrinsic structure without any notable particle-level outward expansion (Supplementary Movie 1), which is confirmed by the similar projected area of AMPSi@C during lithiation process. However, the magnified TEM images (Supplementary Fig. 23) of the pristine state, first lithiated state, and forth lithiated state of AMPSi@C demonstrate pore filling (shown in the red region) and at the same time, the particle size increases and shape changes (shown by the blue arrow). To further study the volume change of the electrode materials, snapshots are taken during lithiation by taking in situ TEM video (Supplementary Movie 2) of a representative Si particle in AMPSi@C at different lithiation time as shown in Supplementary Fig. 24. The size of the partial Si skeleton (shown in red region) is measured to be 105, 112, 117, and 125 nm after lithiation for 3, 20, 40, and 60 s, respectively. Meanwhile, the pore (shown in yellow region) is filled accordingly. The robust structural stability of AMPSi@C is further confirmed by in situ TEM during fast lithiation/delithiation cycling (Figs. 6h–k and Supplementary Movie 2). As shown in Figs. 6h–k, even when a large constant bias of −6/6 V is applied, AMPSi@C shows inward expansion of Si nano skeleton without any notable structural change during four lithiation/delithiation cycles (Supplementary Movie 2). In situ TEM is also performed on AMPSi@C at a higher negative bias (−9 V). As shown in Supplementary Movie 3, the AMPSi@C shows a sudden change leading to abrupt inward expansion of the Si nanoskeleton due to the fast lithiation rate. Nonetheless, the structure of AMPSi@C is sufficiently robust without showing observable mechanical degradation.

The in situ TEM results indicate that AMPSi@C remains stable without cracking and lithiation induced volume expansion of nanoligaments is largely accommodated by the surrounding pores. The inward volume expansion of AMPSi@C enables minimum particle-level outer expansion during lithiation/delithiation cycling, giving rise to small swelling and excellent cycling performance. Moreover, the outer carbon coating enhances electron/ion transport and acts as a protective layer to stabilize SEI on the AMPSi@C electrode resulting in the excellent electrochemical properties.

## Discussion

A vapor dealloying reaction to produce AMPSi from the Mg-Si alloy via a low-cost and scalable top-down approach is designed and described. At 750 °C, Mg reacts with N$_2$ to form liquid Mg$_3$N$_2$[52] and solid Si is separated producing the Mg$_3$N$_2$/Si heterostructure. The in situ generated liquid Mg$_3$N$_2$ acts as the self-template and filler in the Mg$_3$N$_2$/Si hybrid. After removing Mg$_3$N$_2$ in an acidic solution, the bulk Si microparticles are produced, which consist of a bicontinuous porous network and crystalline Si nanoligaments resembling ant nests. The pore size and porosity of AMPSi can be adjusted by varying the Mg concentration in the Mg-Si alloy and nitridation temperature. Moreover, the MgCl$_2$ by-product can be converted into Mg for recycling. This top-down method is simple and economical and can be scaled up for commercial production. In fact, we can produce 3–5 μm AMPSi particles in 110 g per batch in our laboratory using conventional tube furnaces (Supplementary Fig. 2d).

The superior electrochemical properties of AMPSi@C can be attributed to the ant-nest-like structure, which integrates the intrinsic merits of nanoscale and microscale Si. The 3D bicontinuous nanopores enable fast diffusion of the electrolyte and high Li$^+$ accessibility, whereas the interconnected nanoscale Si

ligaments prevent pulverization and cracking. The bicontinuous nanoporous network allows inward volume expansion of Si nanoligaments without obvious particle size change. Our in situ TEM results reveal that the volume change of the Si ligaments is accommodated by the surrounding pores through reversible inward Li breathing without obvious particle size expansion. The as-obtained AMPSi@C has a porosity of 64.3%. The maximum volume accommodation limitation ($\Delta V$) of AMPSi@C is calculated to be 280% according to the equation: $\Delta V = V_{Porosity}/V_{Si} + 1$ without considering binders and conductive additives[17,53]. Here, $V_{Porosity}$ is the pore volume and $V_{Si}$ is the volume of solid Si. The large $\Delta V$ of 280% gives rise to a lithiation capacity of $\sim 2382\,mAh\,g^{-1}$ that is higher than that of AMPSi@C ($2134\,mAh\,g^{-1}$ at 0.1 C) and therefore, in situ TEM demonstrates negligible particle-level outward expansion of AMPSi@C upon lithiation (Supplementary Movie 1–3). The bulk AMPSi@C shows a large tap density and the carbon coating enhances electron/ion transport. Hence, AMPSi@C exhibits enhanced capacity retention and cycling life in comparison with other Si-based anodes. Moreover, inward Li breathing in AMPSi@C gives rise to minimal electrode swelling and large volumetric capacity at the lithiated state. The full cell composed of $Li(Ni_{1/3}Co_{1/3}Mn_{1/3})O_2//$AMPSi@C has a high-energy density of $502\,Wh\,kg^{-1}$ and superior cycle stability with 84% capacity retention after 400 cycles. Owing to self-volume expansion effect of AMPSi@C with negligible particle-level outer expansion, low thickness swelling is achieved in spite of a large areal mass loading and electrode thickness (Supplementary Fig. 21).

In summary, a simple, economical, and scalable nitrogen dealloying technique to fabricate ant-nest-like microscale porous Si from the Mg-Si alloy is reported for the first time. The as-obtained AMPSi@C has continuous pores and interconnected crystalline Si nanoligaments thereby overcoming technical hurdles which have hampered the use of bulk microscale Si in high-performance practical anodes in LIBs. The new design of AMPSi@C simultaneously improves the tap density and electrochemical stability to achieve large volumetric capacity and long-term cycling stability for LIBs. In situ TEM reveals the self-volume inward expansion mechanism of AMPSi@C, which effectively mitigates electrochemically-induced mechanical degradation of the AMPSi@C electrode during cycling and the bulk AMPSi anode shows less than 20% electrode thickness swelling even at a high areal capacity of $5.1\,mAh\,cm^{-2}$. By virtue of these unique structural features, the AMPSi@C electrode shows superior rate capability and long-term cycling stability in full cells with a high-energy density of $502\,Wh\,Kg^{-1}$. Our findings offer insights into the rational design of alloy-based materials that normally undergo large volume changes during operation and application for advanced electrochemical energy storage.

## Methods

### Synthesis of Mg-Si alloy.
The metallurgical Si purchased from Jinzhou Haixin Metal Materials Co., Ltd. was milled to 1–3 μm with a sand mill (Shenzhen Sanxing Feirong Machine Co., Ltd) and then 2.8 g of the milled Si powers were mixed with 5 g of Mg powders (200 mesh, Sinopharm Chemical Reagent Co., Ltd) to form $Mg_2Si$ in a stainless steel reactor heated to 550 °C for 4 h.

### Synthesis of AMPSi and NS-MPSi.
The $Mg_2Si$ powders (3–5 μm) were thermally nitrided in $N_2$ at 750 °C. After thermal reaction, the powders were immersed into 1 M diluted hydrochloric acid to remove $Mg_3N_2$ and the AMPSi powers were collected by filtration. We also prepared nanoparticles assembled 3D mesoporous Si (NS-MPSi) by evaporating Mg from $Mg_2Si$ at 900 °C under vacuum for 6 h as the control sample.

### Synthesis of carbon coating AMPSi (AMPSi@C).
0.4 g of AMPSi and 0.48 g of 2-amino-2-hydroxymethylpropane-1,3-diol (Sigma–Aldrich, 98%) were dispersed in 400 ml of deionized water and 0.6 g of dopamine hydrochloride (Aladdin, 98%) were added under mechanical stirring. The polydopamine-coated AMPSi product

was collected by filtration and then heated at 850 °C for 3 h in $Ar/H_2$. During this process, the polydopamine coating was carbonized into a thin N-doped carbon shell and the final product of AMPSi@C was obtained.

### Materials characterization.
The morphology and microstructure of the AMPSi and AMPSi@C were characterized by field-emission scanning electron microscopy (FE-SEM, FEI Nano 450), transmission electron microscopy (TEM, FEI Titan 60–300 Cs), and high-resolution TEM (HR-TEM, Tecnai G20). The crystal structure, chemical compositions and chemical bonds of materials were characterized by X-ray diffraction (GAXRD, Philips X'Pert Pro), X-ray photoelectron spectroscopy (XPS, ESCALB MK-II, VG Instruments, UK), energy-dispersive X-ray spectroscopy (EDS, Bruker, Super-X), Raman scattering (HB RamLab), and Fourier transform infrared spectroscopy (FTIR, VERTEX 70, Bruker). The surface areas were measured via Brunauer-Emmett-Teller (BET, Micrometrics, ASAP2010) method and the porosity was determined using the mercury porosimeter (Micromeritics, AutoPore V). The 3D structure of AMPSi was obtained by tomographic reconstruction strategy, which was carried out via the total variation based simultaneous algebraic reconstruction technique with synchrotron radiation (National Synchrotron Radiation Laboratory, Hefei, Anhui, China). Thermo-gravimetry (TG, STA449/6/GNETZSCH) was carried out from 30 to 1000 °C at a rate of $5\,°C\,min^{-1}$ in air. The tap density was measured on a vibration density tester (Dandong Haoyu, HY-100B).

### In situ TEM.
The typical nanobattery setup consisted of the working electrode (AMPSi@C), the counter electrode (Li metal), and the solid electrolyte of a naturally grown $Li_2O$ layer. A gold wire was used to capture the AMPSi@C particle by scratching the AMPSi@C sample, which was then transferred and loaded into the nanofactory TEM-scanning tunneling microscope (STM) specimen holder[54]. A tungsten tip was used to scratch Li metal and inserted into a holder in a glove box. Then, the holder was quickly transferred to the TEM instrument (JEOL-2100) and natural thin $Li_2O$ layer was formed on the surface of Li metal due to the native oxidation in air. The $Li_2O$ layer on the tungsten tip was controlled to contact AMPSi@C to complete the nanobattery construction. The Li ions go through the $Li_2O$ layer to alloy with Si at the working electrode under applied voltages of −3, −6, and −9 V.

### Electrochemical tests and electrode swelling measurements.
The electrodes were fabricated via the mixture of active materials, Super-P carbon black, and sodium alginate in water solution to form a slurry at a mass ratio of 8:1:1. The aqueous slurry was coated on a Cu foil by an automatic thick film coater (MTI, MSK-AFA-III) with mass loadings on the electrode of $0.8–2.9\,mg\,cm^{-2}$. After vacuum drying, the electrode with a diameter of 12 mm was prepared with a manual rolling machine. The coin cells (CR2016 type) were assembled in a glove box (Vigor SG1200/750TS-C) by using a Celgard 2400 film as separator, Li foil as a counter electrode, and 1 M $LiPF_6$ in a mixture of diethyl carbonate and ethylene carbonate (1:1) with 6 wt.% VC or FEC additives as the electrolyte. The electrochemical measurements were carried out on the battery tester LAND-CT2001A (Wuhan LAND electronics Co., Ltd., China). Cyclic voltammetry (CV) and electrochemical impedance spectroscopy (EIS) were conducted on a CHI750e electrochemical workstation (Shanghai CH Instrument Company, China). The full cells were assembled with prelithiated AMPSi@C as the anode and commercial $Li(Ni_{1/3}Co_{1/3}Mn_{1/3})O_2$ (NCM) as the cathode. The ratio of negative electrode and positive electrode capacity was about 1.1:1. Galvanostatic charging/discharging was carried out to evaluate the electrochemical performance between 2.8 V and 4.25 V at 0.5 C (1 C = 160 mA $g^{-1}$ based on the cathode active material). Electrochemical prelithiation of AMPSi@C was conducted on the anode of a coin-like half-cell with AMPSi@C as the working electrode and Li foil as the counter electrode. The prelithiation process was conducted via the first discharging process and the working electrode (AMPSi@C) was lithiated to 0.01 V at a 0.05 C rate by a galvanostatic discharging method and this potential was kept for 30 min. After prelithiation, the half-cell of AMPSi@C//Li was disassembled in a glove box and the prelithiated AMPSi@C electrode was taken out quickly, which coupled with the $Li(Ni_{1/3}Co_{1/3}Mn_{1/3})O_2$ cathode to form a full cell.

## Data availability

The data that support the findings of this study are available from the corresponding authors upon reasonable request.

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

## Acknowledgements

This work was financially supported by National Natural Science Foundation of China (No. 51572100, 21875080, 51504171, 21703185, and 61434001), Major Project of Technology Innovation of Hubei Province (2018AAA011), HUST Key Interdisciplinary Team Project (2016JCTD101), Wuhan Yellow Crane Talents Program, and City University of Hong Kong Applied Research Grant (ARG) No. 9667122 and Hong Kong Research Grants Council (RGC) General Research Funds (GRF) No. CityU 11205617. The authors are grateful for the facility support provided by the Nanodevices and Characterization Centre of WNLO-HUST and Analytical and Testing Center of HUST. The authors thank Prof. Yong Guan (National Synchrotron Radiation Laboratory, Hefei, China) and Prof. Yonghui Song (University of Science and Technology of China) for Synchrotron radiation 3D tomographic reconstruction images of the AMPSi and also thank Prof. Qihui Wu (Jimei University) for XPS characterizations.

## Author contributions

K.H. designed the idea and protocol of this work. W.A. and B.G. prepared the materials, carried out the thermogravimetric analysis, XRD, BET and mercury porosimetry and co-wrote the draft. W.A., B.G., J.F. and B.X. conducted the electrochemical tests. S.M. obtained the SEM images. L.W. carried out the TEM characterizations. B.G. and Q.Z. obtained and analyzed the in situ TEM. K.H., W.A., B.G. and P.K.C. discussed the results and co-wrote the manuscript.

## Additional information

**Competing interests:** The authors declare no competing interests.

