## [Peer Review File · Nature Communications]

Reviewers' comments:

Reviewer #1 (Remarks to the Author):

In this manuscript, the authors reported the fabrication of "ant-nest-like" bulk porous silicon/carbon anodes for lithium ion batteries. The result appear to be interesting but the performance is quite modest when compared with other widely reported Si@C based anodes in literature including those of Table S1&2. The analysis of the results are also limited in details and clarity. The work could be considered again after the following major revisions.

(1) Where does the novelty of this contribution reside in terms of its synthesis approach and performance? A balanced chemical reaction equation should be provided for AMPSi conversion reaction. Authors should indicate all synthesis details in Figure 1a if possible and a scale for Figure 1b. In the SI, on material synthesis, what is the purpose of N-doping of the carbon shell? Such doping increases defects and reduces conductivity of carbon. XPS compositional analysis should be provided for the samples to determine the purity as Mg residual particle entrapment is likely with acid induced oxidization.

(2) The authors state that capacities are based on the 'total mass of active materials' This is rather vague. They should directly state what this is based on. Is it the Si content? The Si@C content? The total slurry mass? From a practical standpoint, a total anode mass (minus the current collector) should be used for these calculations to allow direct comparison with other Si anodes.

(3) In Page 3, line 56, it is stated that "Nanostructured Si has a large surface area which reduces the tap density and increases the electrode/electrolyte interfacial area..." Despite the porous nature of AMPSi, the surface area is only 12.6 m²/g with a high tap density of 0.84 g/cm³. This is in contrast with above statement and other reports. Moreover, the authors claimed to have used mercury intrusion porosimetry for porosity analysis but the data are clearly that of nitrogen adsorption-desorption isotherm analysis.

(4) How were the particle size distribution in Figure S2a carried out? It is clear from the TEM/SEM images that the particles are irregular in shape and mostly interconnected, so-called Si nanoligaments (Page 8, line 175), please include error margin in the size analysis. The authors need to provide details of the implications of Figure 3d and not just state the appearance of peaks. What is the chemistry between Si and C in the structure and why it is necessary? The evidence for uniform carbon coating is insufficient in the present results. AMPSi is composed of many single crystals of Si but the interconnected Si framework as a whole is not single-crystal and should be addressed properly and convincingly.

(5) The EIS data should be described in detail with corresponding equivalent circuit. For competitiveness, what is the nature of the performance of this anode at higher C-rates e.g. 5C or 10C? The chemical state evolution of the cycled electrode should be analysed by XPS and compared with the CV results. HRTEM should be provided for Figure S7b to properly illustrate the material stability after cycling.

(6) More details should be provided regarding the characterisation methods stated in the SI. The language in the manuscript also needs polishing.

(7) The authors should give detailed workings for their calculations of the full cell energy density. What masses are included in this?

(8) It is stated that 'The full cells were assembled with prelithiated AMPSi@C (AMPSi@C electrodes were first prelithiated with Li foil and then were taken out for full cell) as the anode and commercial Li(Ni_{1/3}Co_{1/3}Mn_{1/3})O₂ (NCM) as the cathode. A similar strategy was adopted to prepare the cathode electrode.' The authors should give more detail on the electrochemical prelithiation on the anode. What rate was used? what potential/ state of charge or discharge was the anode removed at? Was it a single cycle etc.?The 'similar strategy' for the cathode electrode should also be clarified.

(9) The authors should comment on the role of their electrolyte additive (6% VC), in the SEI formation process. Were alternative electrolyte compositions explored? Additionally, the use of a VC additive

should be mentioned in the main text. The full cell testing was done with an anode limited capacity ratio of 1.1:1. The authors should mention this in the main text as it is important in the context of existing full cell tests.

(10) The in-situ TEM videos shown are not convincing as evidence of cycling of the material. There is no evidence of volume changes or structural changes beyond what would be expected from beam induced effects or variations in diffraction contours as the sample moves slightly. The electrochemical data associated with the TEM experiment should be provided.

Reviewer #2 (Remarks to the Author):

Here are some strengths and weaknesses of the manuscript and its overall suitability for a journal like Nature communications

1. Performance wise AMPSi shows good capacity retention at high mass loading, higher tap density and 80% ICE.
2. The authors have demonstrated full cell data upto 400 cycles with an energy density ~ 500 Wh/Kg.
- 3) Insitu-TEM and post characterization analysis was provided with much details. The authors also provided in some detail how their unique microstructure helps for materials like silicon to accommodate volume expansion.
- 4) Apart from good synthesis strategy good performance was achieved also due to some steps during full cell assemble. Such as prelithiation and forming the SEI in the half cell before making the full cell.

Scope for improvement

1. The manuscript is bit weak on the mechanistic side in explaining why the material shows such great performance. How does exactly the microstructure help to mitigate silicon issues. Insitu TEM work is presented but the full cell work very different from the insitu TEM environment.
2. The authors use Na-alginate as binder. What is the justification of using this compared to for example LI-PAA binders. Does binder play a role into this ?
3. Electrolyte composition - VC is used as an additive but FEC was not used. FEC generally forms a better SEI on silicon surface. This needs to be addressed.
4. Surface functionality- Most silicon surface has a native oxide layer which could be responsible for lower ICE. Even 80% ICE reported is not a good enough in long run although it's better than 70 Or 75% reported in other studies. Surface functionality is one the key factors for SEI composition and stability. The authors should emphasize a few studies on this their work - ACS Appl. Mater. Interfaces., 2014, 6 (10) 7607 and few others. Investigation the surface chemical composition could help to understanding the ICE during the initial cycles.
5. Raman- AMPSi show peaks around 509 cm^{-1} . Its neither amorphous nor crystalline Is? Crystalline or semicrystalline peak should have a peak around 521 cm^{-1} and amorphous Si should be a weak feature around 580 cm^{-1} . This needs to be looked. Even strained Si should not show such large shift compared to crystalline peak at 521 cm^{-1} .

Reviewer #3 (Remarks to the Author):

In this work, a new protocol has been developed to produce high-performance Si-based materials for

Li-ion batteries. The 3D interconnected Si nanoligaments and nanopores of this material prevent its excessive swelling and cracking upon lithiation, resulting in excellent electrochemical performance. The strategy to produce nanostructured/nanoporous Si particles likely to buffer the Si expansion associated with its lithiation is not new. However, the proposed protocol has the advantage to be simple, scalable and yields Si particle with a tap density larger than usual Si nanopowders. The study is well conducted and the obtained electrochemical performances are among the best published to date for Si-based anodes. However, some corrections are required as detailed hereafter.

- It is known that certain variability exists in the electrode capacity measurements for similar experiments (typically around $\pm 5\%$ from my own experiments). Thus, this makes no sense to present the capacity values with one digit after the decimal point, as done through the manuscript.

- line 35: "comprising the AMPSi@C anode" must be replaced by "comprising prelithiated AMPSi@C anode"

- Line 43: the theoretical specific capacity of Si is not 4200 mAh/g ($\text{Li}_{22}\text{Si}_5$) but 3579 mAh/g (i.e. $\text{Li}_{15}\text{Si}_4$) since $\text{Li}_{22}\text{Si}_5$ cannot be formed electrochemically as clearly demonstrated by Dahn et al. several years ago. (see Journal of The Electrochemical Society, 151 (2004) A838-A842).

- It would be relevant to measure the O content in the AMPSi and AMPSi@C powders to confirm that they are not oxidized during or after their synthesis.

- As shown in Fig. 4a, a large capacity decay is observed during the first ~20 cycles for all electrodes. The origin of this decay MUST be discussed. Moreover, this is in contradiction with the CV curves presented in supplementary Fig. 9 suggesting an activation of the electrodes during the first cycles. Actually, it would be more relevant to show the evolution with cycling of selected dQ/dV curves from the Fig 4a experiments.

- The areal mass loading of the electrodes should be indicated in Fig4a-c captions as it has a major impact on the capacity decay as shown in Fig. 4d. The areal capacity of the full-cell presented in Fig. 4e,f and supplementary Fig. 11 must also be indicated

Line 231: the authors indicate a volume capacity of 1712 mAh/cm³ at 0.1C. It must be indicated if this value is determined at the lithiated stage (i.e by considering the volume expansion of the electrode). Information on the cycle number and the areal capacity (or areal mas loading) corresponding to this volume capacity must be also added.

- Lines 238-239: "...these values are the best hitherto reported from Si anodes". This is not true. Larger areal capacities (10 mAh/cm² for +400 cycles) have been obtained by Mazouzi et al. (see Adv. Energy Mater. 2014, 1301718)

- Lines 260-272. The full-cell cycling tests are performed after prelithiation and preactivation of the Si electrodes in half-cells. This procedure is not compatible with battery manufacturing procedures, which prevents the implementation of the present Si electrodes in commercial batteries. This point MUST be discussed in the manuscript. Cycling performance of full-cells obtained with no prelithiated/preactivated Si anodes should be also shown.

- Line 271. The volumetric energy density of the full-cell should be also indicated in addition to the gravimetric energy addition. Actually, the gravimetric energy density is of secondary importance in most consumer applications because the current emphasis is the reduction of device size.

Reply Letter

We thank the reviewers for the in-depth review and valuable comments, which have helped us tremendously to improve the quality and clarity of the paper. We have also conducted additional experiments and provided more discussions to strengthen our claims. All the revisions are highlighted in red in the revised manuscript and supplementary information. We hope that the revisions are satisfactory but if more changes are necessary to improve the paper, please let us know.

Manuscript ID: NCOMMS-18-29967

Reviewers' comments:

Reviewer #1 (Remarks to the Author):

In this manuscript, the authors reported the fabrication of “ant-nest-like” bulk porous silicon/carbon anodes for lithium ion batteries. The result appears to be interesting but the performance is quite modest when compared with other widely reported Si@C based anodes in literature including those of Table 1&2. The analysis of the results is also limited in details and clarity. The work could be considered again after the following major revisions.

Our reply: Thanks for the reviewer’s insightful comments. The encouraging suggestions have been carefully considered and thoroughly addressed as shown below. We appreciate the reviewer’s positive evaluation of the structure design and fabrication of the ant-nest-like microscale porous Si (AMPSi) described in this paper.

Regarding the electrochemical performance of AMPSi@C, we agree that the performance of our AMPSi@C is modest compared to previously reported nano-Si@C electrode materials if we only consider the gravimetric capacity. However, the AMPSi@C shows the large volumetric capacity of 1,760 mAh cm⁻³ at a current density of 420 mA/g, which is highest among Si@C anodes reported so far (see Supplementary Table 1). The AMPSi@C anode with an areal mass loading of 2.9 mg/cm² also delivers a large areal capacity 7.1 mAh cm⁻² at 0.1 mA cm⁻² after 100 cycles and it outperforms most of the reported Si-based anodes (See Supplementary Table 1). Furthermore, the AMPSi@C anode shows very small electrode swelling of 10.1% at an areal capacity of 2.8 mAh cm⁻² close to that of commercial graphite anodes. From the practical standpoint, the volumetric and areal capacity and electrode swelling are critical performance indicators for lithium-ion batteries. In

the revised manuscript, we have added the comparison of the areal and volumetric capacity in Supplementary Table 1 and more discussions. It can be found that our AMPSi@C delivers better electrochemical performance in terms of the tap density and volumetric and areal capacities than these reported Si anode materials.

Supplementary Table 1. Comparison of the electrochemical characteristics and other important features of AMPSi, AMPSi@C, and other reported Si/C composites.

Materials	Content of Si (%)	Tap density (g/cm ³)	Areal mass loading (mg/cm ²)	Cycling (mA h/g)	Rate capacity (mAh/g)	Electrode thickness swelling	Capacity retention of cycles	Areal capacity (mAh/cm ²)	Volumetric capacity (mAh/cm ³)
This work	91.5%	0.8	0.8	1144.3 after 1000 cycles at 0.5 C	847.9 at 2 C 593.4 at 3 C 322.1 at 5 C	10.1%	90%	2.8 at 0.1 mA/cm²	1712 at 0.1 C
	91.5%	0.8	2.9	1614.6 after 100 cycles at 0.25 C	739.7 at 2 C	22.6%	88%	7.1 at 0.1 mA/cm²	1760 at 0.1 C
Pomegranate Si/C (Ref.20)	77%	0.53	3.12	1,160 after 1,000 cycles at 0.5 C	690 at 2 C	-	97%	3.7 at 0.03 mA/cm ²	1270 at C/20
Micro-sized Si-C composite (Ref.11)	80%	0.78	1.2	1200 after 600 cycles at 0.3 C	990 at 1.5 C	44%	96%	1.9 at 0.3 mA/cm ²	1326 at 0.1 C
Micrometer-sized porous Si (Ref.50)	82.7%	0.72	2.11	1467 after 370 cycles at 0.62 C	650 at 2.6 C	-	83%	2.8 at 0.2 mA/cm ²	1075 at 0.06 C
Fe-Cu-Si composite (Ref.15)	~70%	0.8	0.8	420 after 50 cycles at 0.5 C	429 at 1.2 C	49%	90%	3.44 at 0.2 mA/cm ²	1030 at 0.05 C
Porous Si sponge (Ref.27)	40%	-	0.5	570 after 1,000 cycles at 0.25 C	410 at 1 C	30%	80%	1.8 at 0.06 mA/cm ²	-
Watermelon-Inspired Si/C Microsphere (Ref.6)	12.5 %	0.68	4.1	450 after 250 cycles at 0.5 C	~500 at 5 C	-	80%	2.54 at 0.2 mA/cm ²	~420 at 0.07 C
Si/N-doped C /CNT (Ref.14)	71%	-	0.8	1031 after 100 cycles at 0.15 C	~600 at 0.5 C	-	67%	1.07 at 0.42 mA/cm ²	-
Si cube (Ref.26)	81.4%	0.25	1.0	1338 after 200 cycles at 0.48 C	907 at 2.4 C	41.7%	77.6%	1.9 at 0.2 mA/cm ²	~480 at 0.05 C
Porous coral-like Si (Ref.28)	-	-	~1.0	1956 after 100 cycles at 0.01 C	971 at 2 C	65%	79.8%	~2.0 at 0.1 mA/cm ²	-
Embedded Graphitic C Shell on Si (Ref.51)	82%	-	0.9	1056 after 800 cycles at 0.48 C	1155 at 1.4 C	-	~66%	1.9 at 0.2 mA/cm ²	-

1) (1) *Where does the novelty of this contribution reside in terms of its synthesis approach and performance?* (2) *A balanced chemical reaction equation should be provided for AMPSi conversion reaction. Authors should indicate all synthesis details in Fig. 1a if possible and a scale for Fig. 1b.* (3) *In the SI, on material synthesis, what is the purpose of N-doping of the carbon shell? Such doping increases defects and reduces conductivity of carbon.* (4) *XPS compositional analysis should be provided for the samples to determine the purity as Mg residual particle entrapment is likely with acid induced oxidization.*

Our reply: Thanks for the reviewer's helpful comments. Below is our point-to-point response to the reviewer's comments numbered above.

(1) The novelty of this contribution includes the structure design, facile and economic synthesis, as well as superior electrochemical performance. Firstly, this paper reports for the first time the ant-nest-like microscale porous Si (AMPSi) consisting of 3D interconnected Si nanoligaments and continuous nanopores. This unique ant-nest-like materials are fundamentally different from previously reported Si/C composite secondary particles. To characterize this novel structure, we conduct synchrotron radiation tomographic reconstruction of the AMPSi (Fig. 2e in the revised manuscript and Fig. R1 in the reply letter) and the results clearly confirm the 3D continuous nanopore structure and interconnected Si nanoligaments. Secondly, this paper describes a novel top-down technique to produce AMPSi from commercial Si or Mg-Si alloy powder *via* simple nitridation under nitrogen followed by removal of the Mg₃N₂ by-product in acid solution. This strategy that is cost effective, environmentally friendly, and easy to scale production has large commercial potential for high energy lithium-ion batteries. Thirdly, the AMPSi integrates the inherent merits of nanoscale and microscale Si. The nanoligaments effectively prevents pulverization and the well-defined continuous nanopores accommodate the volume change of Si nanoligaments through reversible inward Li breathing. Therefore, the AMPSi@C anode with an areal mass loading of 2.9 mg cm⁻² shows a small volume change upon lithiation and the highest volumetric capacity of 1,760 mAh cm⁻³ at 0.1 C (1 C = 4.2 A/g) among Si@C-based anodes reported so far. Moreover, the AMPSi@C with a high areal mass loading of 2.9 mg cm⁻² delivers a large areal capacity of 7.1 mAh cm⁻² at 0.1 mA cm⁻² and shows excellent cycle stability with 0.015% decay per cycle for over 1000 cycles, which outperform most of the reported

Si-based anodes (See Supplementary Table 1 and 2). The novel nanostructure, facile and scalable production strategy, as well as superior battery performance are the main novelties of this work.

Fig. R1. The 3D structure of AMPSi obtained by the synchrotron radiation tomographic reconstruction with the total variation (TV)-based simultaneous algebraic reconstruction technique.

(2) A balanced chemical reaction equation has been provided for AMPSi conversion reaction in the revised text.

We also provide the synthesis details in Fig. 1a in the revised manuscript. The scale bar for Fig. 1b has been added to the revised text.

(3) To improve the conductivity, we coat AMPSi with carbon layers. Dopamine is chosen as the carbon precursor due to the high carbonization yield (60%) and strong and versatile anchoring capability on Si^{1-3} . In addition, dopamine can self-polymerize *in situ* to produce a highly conformal polydopamine (PDA) coating with tunable thickness. Therefore, the carbon shell thickness can be easily controlled by adjusting the concentration of dopamine and polymerization time. Since dopamine contains N, the shell derived from the carbonized PDA coating contains N as well. We agree that the N doping increases defects and may decrease the initial Coulombic efficiency. However, the N content in carbon shell derived from PDA coating at 850 °C carbonization temperature is less than 4%.⁴ Moreover, previous reports indicate that N can introduce donor states near the Fermi level to generate enhanced n-type conductivity⁵⁻⁹.

(4) To confirm if the Mg is removed completely, we have conducted XPS analysis of AMPSi (Supplementary Fig. 3) and no Mg peak is observed confirming complete removal of Mg during synthesis (Fig. R2 in this reply letter).

Fig. R2. XPS survey (a) and high-resolution spectrum of Mg_{1s} (b) of AMPSi.

2) *The authors state that capacities are based on the ‘total mass of active materials’ This is rather vague. They should directly state what this is based on. Is it the Si content? The Si@C content? The total slurry mass? From a practical standpoint, a total anode mass (minus the current collector) should be used for these calculations to allow direct comparison with other Si anodes.*

Our reply: We thank the reviewer for this helpful suggestion. We apologize for the confusion of the description of the capacities that are based on the ‘total mass of active materials’. In our work, the capacities of AMPSi@C are calculated based on the total mass of AMPSi and C coating. Only AMPSi is based on the AMPSi mass loading. In our revised manuscript, we have added a sentence “All the specific capacity values in this paper are based on the total mass of AMPSi@C, unless otherwise stated”.

Indeed, when calculating the specific capacity, both the redox active and inactive materials in the electrodes such as the binder, conductive additives, and current collector must be considered from a practical standpoint. However, in most papers on Si anodes, the capacities are calculated based on the mass of active materials such as Si or Si@C, not including the binders, conductive additives, and electrolytes. Therefore, we calculate the capacities based on the mass of Si@C in order to compare with reported results of Si anode materials (Supplementary Table 1). In the revision, we also provide the capacity of AMPSi@C based on the total mass including the

active materials, binder, and conductive additive (minus the current collector) in Supplementary Table 3.

Supplementary Table 3. Comparison of the capacities based on the total mass including active material, binder and conductive additive (minus the current collector) between AMPSi@C and other reported Si-based anodes

Materials	Mass ratio of active materials, carbon black and binder (%)	Cycling capacity based on total materials mass (mA h/g)	Rate capacity based on total materials mass (mAh/g)
This work	80:10:10	915.4 after 1000 cycles at 0.5 C 1291.7 after 100 cycles at 0.25 C	678.4 at 2 C 474.7 at 3 C
Pomegranate Si/C (Ref.20)	80:10:10	928 after 1,000 cycles at 0.5 C	552 at 2 C
Micro-sized Si-C composite (Ref.11)	60:20:20	875.4 after 200 cycles at 0.24 C	594 at 1.5 C
Micrometer-sized porous Si (Ref.50)	70:10:20	880 after 370 cycles at 0.62 C	455 at 2.6 C
Fe-Cu-Si composite (Ref.15)	80:10:10	336 after 50 cycles at 0.5 C	343 at 1.2 C
Porous Si sponge (Ref.27)	40:40:20	228 after 1,000 cycles at 0.25 C	164 at 1 C
Watermelon-Inspired Si/C Microsphere (Ref.6)	90:5:5	405 after 250 cycles at 0.5 C	~450 at 5 C
Si/N-doped C /CNT (Ref.14)	65:20:15	670 after 100 cycles at 0.15 C	~390 at 0.5 C
Si submicrocube (Ref.26)	80:10:10	1070 after 200 cycles at 0.48 C	725.6 at 2.4 C
Porous coral-like Si (Ref.28)	70:20:10	1369 after 100 cycles at 0.01 C	680 at 2 C

3) In Page 3, line 56, it is stated that “Nanostructured Si has a large surface area which reduces the tap density and increases the electrode/electrolyte interfacial area...” Despite the porous nature of AMPSi, the surface area is only 12.6 m²/g with a high tap density of 0.84 g/cm³. This is in contrast with above statement and other reports. Moreover, the authors claimed to have used mercury intrusion porosimetry for porosity analysis but the data are clearly that of nitrogen adsorption-desorption isotherm analysis

Our reply: Thanks for the reviewer's comment. The tap density=mass (g) /volume (cm³) and the volume is the volume of the powder sample mechanically vibrated in a container including the particle volume and void volume among the Si particles. Nanostructured Si has a large pore volume and void volume among the Si nanoparticles, thereby resulting in a small tap density and low volumetric capacity. On the other hand, the large surface area of nanostructured Si gives rise to a large electrode/electrolyte interfacial area resulting in low initial coulombic efficiency when used as active materials in electrodes. Compared to nanosized Si, microsized Si generally has a larger tap density. The AMPSi particles have the size of 3-5 μm but contain continuous nanopores. However, due to the space-efficient packing of AMPSi, its tap density is significantly higher than that of primary nanosized particles packed randomly. The measured tap density of AMPSi is as high as 0.84 g/cm³ which is much larger than that of nanostructured Si (generally less than 0.4 g/cm³).

Since the well-organized pores are continuous and most pores are larger than 50 nm, we use mercury intrusion porosimetry to evaluate the porosity of the large pores (larger than 50 nm) and nitrogen adsorption-desorption isotherms to assess the mesopores. The BET surface area of AMPSi is measured to be 12.6 m²/g. AMPSi integrates the merits of microscale Si and nanoscale Si, thereby exhibiting large tap density, low BET surface area and high structural stability upon lithiation/delithiation. In the revision, we have added the discussion on the tap density and surface area as highlighted in red in the revised manuscript.

4) (1) *How were the particle size distribution in Fig. S2a carried out? It is clear from the TEM/SEM images that the particles are irregular in shape and mostly interconnected, so-called Si nanoligaments (Page 8, line 175), please include error margin in the size analysis.* (2) *The authors need to provide details of the implications of Fig. 3d and not just state the appearance of peaks.* (3) *What is the chemistry between Si and C in the structure and why it is necessary?* (4) *The evidence for uniform carbon coating is insufficient in the present results.* (5) *AMPSi is composed of many single crystals of Si but the interconnected Si framework as a whole is not single-crystal and should be addressed properly and convincingly.*

Our reply: We thank the reviewer for this constructive suggestion. Below is our point-to-point response to the reviewer's comments numbered above.

(1) The particle size distribution is evaluated by a laser particle size analyzer

(Mastersizer 2000). The laser particle size analyzer measures the intensity of light scattered as a laser beam passes through a dispersed particulate sample and the data are analyzed to calculate the size of the particles that create the scattering pattern. Indeed, the particles are irregular in shape for AMPSi according to the TEM/SEM images, however, the laser particle size analyzer shows an equivalent spherical diameter based on the same volume for the irregular particle¹⁰ and D_{50} (The percentage of particles of this size in all particles is 50%) is generally used to evaluate the average diameter. The measured D_{50} values of AMPSi are 3.0, 3.2, 3.0, 3.0 and 2.9 for five similar samples and thus the error margin of D_{50} is ± 0.2 . In this revision, we have added the D_{50} result and the error margin as highlighted in red in the revised manuscript.

(2) We have added details of the implication of Fig. 3d to the revised manuscript. Raman scattering (Fig. 3d) of AMPSi shows a sharp peak at 510 cm^{-1} and two weak peaks at 299 and 925 cm^{-1} , corresponding to the characteristic peaks of crystalline Si.^{3,11} The three peaks are still clearly visible in AMPSi@C but the strong peak blue-shifts to 504 cm^{-1} possibly due to phonon confinement as a result of the conformal carbon coating on AMPSi.^{12,13} The two vibration peaks at 1347 and 1581 cm^{-1} are assigned to the vibration modes of disordered graphite (D band) and E_{2g} of crystalline graphite (G band) and the large I_D/I_G ratio (1.14) reflects the low graphitic degree of the carbon coating in accordance with the XRD results of amorphous carbon (Supplementary Fig. 8). In the revision, we have added the discussion on Fig. 3d as highlighted in red.

(3) To reveal the chemistry between Si and C in the AMPSi@C structure, we have acquired Fourier transform infrared (FTIR) spectra from AMPSi and AMPSi@C (Supplementary Fig. 7 and Fig. R3 in the reply letter). The bands at 1060 and 1620 cm^{-1} correspond to the characteristic vibrations of Si and compared to AMPSi, the band at 1060 cm^{-1} is split into the two peaks at 1240 and 1090 cm^{-1} in AMPSi@C, suggesting robust bonding at the interfaces between Si and coated carbon shell.³ The strong bonding between carbon and Si improves electron transfer at the interface and stabilizes the SEI to enhance the electrochemical performance.

Fig.R3 FTIR spectra of the AMPSi and AMPSi@C.

(4) In our work, dopamine is used as the carbon precursor of carbon shell which can self-polymerize into polydopamine (PDA) films immobilized on Si with robust adhesion. Thus, complete and homogeneous carbon coatings with strong bonding with Si can be achieved after carbonization.¹⁻³ To confirm the uniform carbon coating, we have conducted HR-TEM on AMPSi@C at different regions. The HR-TEM images and EDS maps in Supplementary Fig. 6 (Fig. R4 in the reply letter) reveal the uniform carbon shell on the surface of Si in AMPSi@C. The results confirm the formation of a uniform C coating on the AMPSi and we have added the discussion in the text as highlighted in red.

Fig. R4. HR-TEM images (a and b) of AMPSi@C and EDS mapping (c), revealing uniform C are coated on AMPSi with the thickness of 5-8 nm. The red and yellow correspond to Si and C elements, respectively.

(5) We apologize for this confusion. HR-TEM and selected-area electron diffraction (Supplementary Fig. 4) reveal that AMPSi is composed of many single crystals of Si but the interconnected Si framework as a whole is not single-crystal. In this revision, we have revised the related description as shown in red.

5) (1) The EIS data should be described in detail with corresponding equivalent circuit. (2) For competitiveness, what is the nature of the performance of this anode at higher C-rates e.g. 5C or 10C? (3) The chemical state evolution of the cycled electrode should be analyzed by XPS and compared with the CV results. (4) HRTEM should be provided for Fig. S7b to properly illustrate the material instability after cycling.

Our reply: Thanks for the reviewer's valuable suggestions. Below is our point-to-point response to the reviewer's comments numbered above.

(1) In this revision, we provide the corresponding equivalent circuit of EIS data in the Supplementary Fig. 18 (Fig. R5 in the reply letter). R_{Ω} is the internal resistance including the electrolyte resistance and contacting resistance. The depressed semicircles in the high-frequency region represent the overlap of the resistance of the SEI film (R_{sf}) and charge transfer resistance (R_{ct}).^{14,15} It is found that R_{sf} of AMPSi@C is smaller than that of AMPSi, meaning that the SEI at AMPSi@C is thinner and more stable than that of AMPSi. Moreover, R_{ct} of AMPSi@C is lower than that of AMPSi, suggesting smaller charge transfer resistance in AMPSi@C due to the enhanced conductivity of the carbon coating. In this revision, the EIS data and corresponding equivalent circuit are added and discussed as highlighted in red.

Fig. R5. The Nyquist plots (a) of the AMPSi and AMPSi@C and corresponding equivalent circuit (b).

(2) Following the reviewer's suggestion, we have assessed the electrochemical performance of AMPSi@C at a higher density of 5.0 C (21,000 mA g⁻¹) as Supplementary Fig. 14 (Fig. R6 in the reply letter). At a large density of 5.0 C, AMPSi@C still shows good cycling stability with a capacity decay of 0.08% per cycling for over 500 cycles thus outperforming most of the reported Si-based anode materials (See Supplementary Table 1). In the revision, the electrochemical performance at a higher density of 5.0 C is added and highlighted in red.

Fig. R6. Long-term cycling test conducted at 0.5 C in the first three cycles and at a larger current density of 5 C in the later 1,000 cycles (1 C = 4,200 mA g⁻¹).

(3) The chemical states of the cycled electrode are analyzed by *ex situ* XPS (Supplementary Fig. 17 and Fig. R7 in the reply letter). Cyclic voltammetry (Supplementary Fig. 16a) of AMPSi@C show the typical lithiation and delithiation peaks. The peak at 0.16 V in the cathodic branch (lithiation) is assigned to the conversion of Si to Li_xSi and in the anodic branch (delithiation), the two peaks at 0.40 V and 0.53 V are attributed to delithiation of Li_xSi back to amorphous Si. Hence, we select pristine AMPSi@C and the three samples at different lithiated (0.1 V) and delithiated (0.4 and 0.53 V) potentials during the 1st cycle for the *ex situ* XPS analyses. To prepare the samples, the half cells at different charging and discharging states are disassembled in glove box and the electrodes are taken out. The residual electrolytes and organic SEI are removed by soaking the electrodes in acetonitrile, washed with ethanol and methanol, and then kept in an argon-filled dry box. Before conducting XPS characterization, the AMPSi@C electrodes are etched by Ar⁺ about 10 nm to remove the inorganic SEI and surface oxide.

The fine XPS results of Si 2p at different lithiated (0.1 V) and delithiated (0.4 and 0.53 V) potentials during the 1st cycling are shown in Supplementary Fig. 18 (Fig. R7

in this reply letter). The pristine AMPSi@C exhibits two Si peaks at 98.8 and 99.4 eV assigned to Si 2p_{1/2} and Si 2p_{3/2} of elemental Si (Si⁰). After full lithiation to 0.01 V, the two Si peaks shift to 97.3 and 97.9 eV due to the alloying reaction to form the Li_xSi phase. When the electrode is delithiated at 0.40 V, the peaks of Si⁰ reappear and those corresponding to Li_xSi shift to high binding energies suggesting partial Li-ion desertion from the Li_xSi alloy. At the higher delithiation voltage of 0.53 V, stronger peaks of Si⁰ are observed in line with the Si binding energy in Li_xSi shifting to higher energy, meaning decrease of the Li content in Li_xSi due to more Li-ion desertion from Li_xSi alloy. The *ex situ* XPS results agree well with the CV results, confirming the alloying/delithiation reactions of AMPSi@C during lithiation/delithiation. In the revision, we have added the discussion of the *ex situ* XPS as highlighted in red.

Fig.R7. The fine XPS Si 2p results of the lithiated (0.01 V) and delithiated (0.40 and 0.53 V) samples during the 1st cycle.

(4) In this revision, we provide the HR-TEM image of the nanoparticles assembled 3D mesoporous Si (NS-MPSi) after 600 cycles as Supplementary Fig. 11 (Fig. R8 in this reply letter) and added the discussion. The HR-TEM image reveals that the structure of NS-MPSi is destroyed after 600 cycles resulting in the fast capacity attenuation as shown in Fig. 4a.

Fig. R8. HRTEM image of NS-MPSi after 600 cycles.

6) *More details should be provided regarding the characterization methods stated in the SI. The language in the manuscript also needs polishing.*

Our reply: We thank the reviewer for the good comment. We have added the detailed materials characterization methods in the revised manuscript. Besides, we have double-checked the English to polish the language throughout the manuscript.

7) *The authomngbairs should give detailed workings for their calculations of the full cell energy density. What masses are included in this?*

Our reply: We appreciate the reviewer's valuable suggestions. The energy density (E) is determined by the electrochemical voltage (U) of the battery and the specific capacity (C) of the electroactive materials in the electrodes. The voltage window (U) describes the potential range from the charging upper limit to the discharging lower limit. The average voltage (U) of the full cell is obtained by the incremental capacity curve (dQ vs. V) (Fig. R9 in the reply letter) based on the differential of the discharging curve and the measured average voltage (U) is 3.75 V. The capacity of the battery is calculated based on the total mass of active materials of both $\text{Li}(\text{Ni}_{1/3}\text{Co}_{1/3}\text{Mn}_{1/3})\text{O}_2$ cathode and AMPSi@C anode and the discharging capacity of the full cell is 134 mAh g^{-1} . Therefore, the energy density $E = U \times C = 3.75\text{V} \times 134 \text{mAh g}^{-1} = 502 \text{Wh kg}^{-1}$. This value is only indicative since it only includes the mass of both active materials of the anode and the cathode and does not contain the weight of the battery ancillary components such as the electrolyte, binder and current collectors^{16,17}. In the revision, we have added details about the calculation of the full cell energy density.

Fig. R9. (a) Discharging capacity vs. voltage (V) curves and (b) dQ vs. V curves of the AMPSi@C//NCM full cell.

8) *It is stated that ‘The full cells were assembled with prelithiated AMPSi@C (AMPSi@C electrodes were first prelithiated with Li foil and then were taken out for full cell) as the anode and commercial $\text{Li}(\text{Ni}_{1/3}\text{Co}_{1/3}\text{Mn}_{1/3})\text{O}_2$ (NCM) as the cathode. A similar strategy was adopted to prepare the cathode electrode.’ The authors should give more detail on the electrochemical prelithiation on the anode. What rate was used? what potential/ state of charge or discharge was the anode removed at? Was it a single cycle etc.? The ‘similar strategy’ for the cathode electrode should also be clarified.*

Our reply: We thank the reviewer for the valuable comments. The electrochemical prelithiation on the anode is conducted on a coin-like half-cell with AMPSi@C as the working electrode and Li foil as the counter electrode. The electrolyte is 1.0 M LiPF_6 in 1:1 v/v ethylene carbonate/diethyl carbonate containing 6 vol % vinylene carbonate. The prelithiation process is only conducted *via* the 1st discharging process and the working electrode (AMPSi@C) is lithiated to 0.01 V at a 0.05 C rate by a galvanostatic discharging method and the potential is kept for 30 min. After prelithiation, the half-cell of AMPSi@C//Li is disassembled in a Ar-filled glove box and the AMPSi@C electrode is quickly taken out and assembled with the $\text{Li}(\text{Ni}_{1/3}\text{Co}_{1/3}\text{Mn}_{1/3})\text{O}_2$ (NCM) cathode to form a full cell. In this revision, we have provided the details of prelithiation of AMPSi@C in the Methods section in revised text as highlighted in red.

As for the cathode, the commercial $\text{Li}(\text{Ni}_{1/3}\text{Co}_{1/3}\text{Mn}_{1/3})\text{O}_2$ (NCM) is directly used without prelithiation. We are sorry for the inaccurate statement and confusion. In the revised text, we have deleted the sentence "A similar strategy was adopted to prepare the cathode electrode".

9) *The authors should comment on the role of their electrolyte additive (6% VC), in the SEI formation process. Were alternative electrolyte compositions explored? Additionally, the use of a VC additive should be mentioned in the main text. The full cell testing was done with an anode limited capacity ratio of 1.1:1. The authors should mention this in the main text as it is important in the context of existing full cell tests.*

Our reply: Thanks for the reviewer's helpful suggestion. We have not explored alternative electrolyte compositions in this paper. The electrolyte employed in our work is 1.0 M LiPF₆ in 1:1 v/v ethylene carbonate and diethyl carbonate (EC-EDC) containing 6 vol % vinylene carbonate (VC). VC is widely used as an electrolyte additive to stabilize the solid electrolyte interphase (SEI) on Si to improve capacity retention and enhance the thermal stability of the lithiated Si anode¹⁸. In the revision, we have added the discussion on the role of VC additive in the SEI formation process. Regarding the full cell testing, we added a sentence: "*The full cell test is done with an anode limited capacity ratio of 1.1:1 in order to consider safety and capacity matching of the full cell during cycling*".

10) *The in-situ TEM videos shown are not convincing as evidence of cycling of the material. There is no evidence of volume changes or structural changes beyond what would be expected from beam induced effects or variations in diffraction contours as the sample moves slightly. The electrochemical data associated with the TEM experiment should be provided*

Our reply: Thanks for the reviewer's constructive suggestion. The reason why the AMPSi@C has negligible particle-level outward expansion or structural changes in the *in situ* TEM videos is due to the Si nanoligaments expand through reversible inward Li breathing and inward void filling. The magnified TEM images (Supplementary Fig. 23) of the pristine state, and first and fourth lithiated states of AMPSi@C clearly demonstrate pore filling (shown in red region) and concurring the particle size increases and particle shape changes (shown by the blue arrow in Fig. R10). To further investigate the volume change of the electrode materials, we provide the TEM snapshots during lithiation by the *in situ* TEM video (Supplementary Video 2 of our revised manuscript) taken from a representative particle in AMPSi@C at different lithiation time as shown in Supplementary Fig. 24

(Fig. R11 in this reply letter). The size of the partial Si skeleton (shown in red region) is measured to be 105, 112, 117, 125 nm after lithiation for 3, 20, 40, 60 s, respectively. Meanwhile, the pore (shown in yellow region) is filled accordingly. The results clearly confirm that the Si size increases and pore filling during lithiation although the structure of AMPSi@C remains stable. Moreover, we have carried out *in situ* TEM on AMPSi@C at a higher negative bias (-9 V), as shown in Supplementary Video 3. AMPSi@C shows a sudden change leading to an abrupt inward expansion of Si nanoskeleton due to the fast lithiation rate. Nonetheless, the structure of AMPSi@C is sufficiently robust and no mechanical degradation is observed further confirming the excellent structural ability.

To avoid the electron beam effect in our *in situ* TEM experiments¹⁹, our *in situ* TEM is conducted with an electron dose below the safe dose of 1 A cm⁻². Thus, the effects of the electron beam to induce chemical lithiation or electrochemical lithiation are suppressed as confirmed by other groups^{20,21}.

We conduct *in situ* TEM on the solid-state open cell by applying a bias (not current/current density) to lithiated/delithated the electrode. AMPSi@C is the working electrode, Li metal is the counter electrode, and native Li₂O layer is the solid electrolyte. When a negative potential is applied to the AMPSi@C with respect to the counter electrode of Li metal, Li begins to diffuse into one end of the AMPSi@C through the Li₂O layer. In this process, the thickness of the native Li₂O layer and contact between AMPSi@C and Li/Li₂O affect the current of the open cell. Meanwhile, the observed current is rather small (normally lower than 1nA) making it hard to measure. Moreover, the *in situ* TEM observation does not consider the effect of the electrolytes and solid electrolyte interphase (SEI) as typically observed in LIBs and thus, it is difficult to conduct lithiation/delithiation during *in situ* TEM by controlling the current/current density and then to record the real charge/discharge curves. However, it is generally accepted that the materials behavior observed by *in situ* TEM using the open cell design reflects the real battery electrochemistry in terms of the structural and chemical evolution of electrodes upon lithiation/delithiation^{19,22}. In fact, in the past few years, *in situ* TEM studies of LIBs have been carried out based on the open cell and provide important fundamental information about the reaction kinetics and microstructural evolution during battery operations in real time^{19,21}. For example, the recently developed “operando” TEM electrochemical liquid cell consisting of the configuration of a real battery with a relevant liquid electrolyte

indicates that the structural and chemical evolution of Si nanowires in both the open cell design and closed cell designs being similar except for the formation of a thick SEI layer on the nanowires^{19,22,23}. Liu et al. have reported anisotropic swelling of Si nanowires during lithiation by using the open cell in TEM²⁴. Anisotropic lithiation of Si nanopillars and microslabs have been reported by other groups using *ex situ* experiments based on real LIBs^{25,26}. The consistency shows that the electrochemical-mechanical response observed in *in situ* TEM is intrinsic to the electrode materials and the information obtained here on AMPSi@C is comparable to that of real electrochemical cells.

Fig. R10. *In situ* TEM observations of AMPSi@C of pristine (a), the first lithiated (b), and the fourth lithiated (c) of AMPSi@C. The red dashed boxes indicate the pore filling during lithiation and concurring the particle size increases and particle shape changes (shown by the blue arrow).

Fig. R11. Time-resolved TEM images of AMPSi@C at different times taken from *in-situ* TEM test. The size of the partial Si skeleton (shown in red region) is measured to be 105, 112, 117, 125 nm after lithiation for 3, 20, 40, 60 s, respectively. Meanwhile, the pore (shown in yellow region) is filled accordingly.

Reviewer #2 (Remarks to the Author):

Here are some strengths and weaknesses of the manuscript and its overall suitability for a journal like Nature communications

- 1. Performance wise AMPSi shows good capacity retention at high mass loading, higher tap density and 80% ICE.*
- 2. The authors have demonstrated full cell data up to 400 cycles with an energy density ~ 500 Wh/Kg.*
- 3) In situ-TEM and post characterization analysis was provided with much details. The authors also provided in some detail how their unique microstructure helps for materials like silicon to accommodate volume expansion.*
- 4) Apart from good synthesis strategy good performance was achieved also due to some steps during full cell assemble. Such as prelithiation and forming the SEI in the half cell before making the full cell.*

Our reply: We appreciate the reviewer's positive comments. The referee's encouraging comments have been carefully considered and thoroughly addressed as shown below.

Scope for improvement

- 1) The manuscript is bit weak on the mechanistic side in explaining why the material shows such great performance. How does exactly the microstructure help to mitigate silicon issues. In situ TEM work is presented but the full cell work very different from the in situ TEM environment.*

Our reply: We thank the reviewer for these constructive suggestions. We have conducted synchrotron radiation tomographic reconstruction of the AMPSi and the images are provided as Fig. 2e (Fig. R1 in this reply letter). Different from the previously reported porous Si, the synchrotron radiation tomographic images clearly confirm that AMPSi consists of 3D interconnected Si nanoligaments and continuous nanopores. The 3D interconnected nanoligaments of AMPSi effectively suppress the stress to maintain the high structural stability and avoid pulverization upon lithiation,

while the continuous nanopores allow inward expansion upon lithiation thereby enabling negligible particle-level outward expansion. Moreover, the continuous nanopores are favorable to electrolyte diffusion and Li-ion transport. These remarkable features of AMPSi are confirmed by *in situ* TEM observation during alloying/dealloying.

We agree with the reviewer that the full cell works differently compared to that in the *in situ* TEM environment and understand that the *in situ* results discussed in the paper do not consider the effect of the electrolytes and solid electrolyte interphase (SEI) as typically observed in LIBs. However, it is generally accepted that the materials behaviors observed by *in situ* TEM using the open cell design reflects the real battery electrochemistry in terms of the structural and chemical evolution of the electrodes upon lithiation/delithiation^{19,22}. In fact, over the past few years, *in situ* TEM has been conducted on LIBs using the open cell and provided important fundamental knowledge about the reaction kinetics and microstructural evolution during battery operation in real time^{19,21}. For example, the recently developed “operando” TEM electrochemical liquid cell consisting of the configuration of a real battery with a relevant liquid electrolyte indicates that the structural and chemical evolution of Si nanowires in both the open cell design and closed cell designs being similar except for the formation of a thick SEI layer on the nanowire^{19,22,23}. This consistency shows that the electrochemical-mechanical response in *in situ* TEM is intrinsic to the electrode materials and the electrochemical-mechanical responses observed by *in situ* TEM are intrinsic to the electrode materials. Therefore, the information obtained from the *in situ* TEM studies on AMPSi@C is comparable to that of real electrochemical cells.

In the revision, we have added synchrotron radiation tomographic images and discussion to explain why the AMPSi@C shows good mechanical stability and Li storage performance in the revised text, which are highlighted in red.

Fig. R1. 3D structure of AMPSi obtained by the synchrotron radiation tomographic reconstruction of the projections which was carried out by the total variation (TV)-based simultaneous algebraic reconstruction technique.

2) *The authors use Na-alginate as binder. What is the justification of using this compared to for example LI-PAA binders? Does binder play a role into this?*

Our reply: Thanks for the reviewer's helpful comments. Sodium alginate (Na-alginate) is widely used as a water-soluble binder for electrode materials. Na-alginate contains a higher content of carboxylic group on each monomeric unit enabling a great number of hydrogen bonds between the binder and native oxide layer of the Si active materials²⁷⁻²⁹. The high binding ability of Na-alginate results in better electrode stability and adhesion ability compared to the conventional polyvinylidene fluoride (PVDF) binder²⁷⁻²⁹. Therefore, we use Na-alginate as the binder in our work. We have not investigated the influences of different binders on the electrochemical performances of AMPSi@C anode materials. In our control experiment, we evaluate the Li-ion storage performance of nanoparticles assembled 3D mesoporous Si (NS-MPSi) using the Na-alginate as binder but NS-MPSi shows poor cycle stability and lower capacity compared to AMPSi@C. Therefore, we believe the intrinsic features of AMPSi@C play important roles in efficient Li storage.

3) *Electrolyte composition - VC is used as an additive but FEC was not used. FEC generally forms a better SEI on silicon surface. This needs to be addressed.*

Our reply: We thank the reviewer for the constructive suggestion. Fluoroethylene carbonate (FEC) and vinylene carbonate (VC) are the two widely used additives in both Li- and Na-ion batteries. Both additives are structurally related to ethylene carbonate (EC). FEC is a fluorinated form of ethylene carbonate (EC), whereas VC

contains a double bond which can polymerize during decomposition to form oligomers and insoluble polymers¹⁸. Previous reports have demonstrated that both FEC and VC can improve the capacity retention of Si anodes by forming highly cross-linked PEO/ poly (VC) in SEI, thereby resulting in positive implication for the stability of Si-based anodes¹⁸. It is generally accepted that FEC forms a better SEI on silicon but a recent paper suggests that VC additive results in higher average Coulombic efficiency (CE)¹⁸. In the reversion, we compare the electrochemical performance of AMPSi@C with FEC or VC as electrolyte additives (Supplementary Figs. 15 a-b). Although the AMPSi@C anodes display similar cycle stability, the AMPSi@C anode using VC as additive exhibits slightly larger capacities from the 2nd to 50th cycles (Fig. R12 in this reply letter). Moreover, the AMPSi@C anode with VC as additive shows a slightly higher average CE compared to FEC additive (Fig. R13 in this reply letter). In the revision, we provide the electrochemical properties of FEC or VC as electrolyte additives (Supplementary Figs. 15 a-b) and added the description as highlighted in red color.

Fig. R12. Cycle performances of AMPSi@C electrodes in two electrolyte solutions with VC and FEC as additives

Fig. R13. Coulombic efficiency of AMPSi@C electrodes upon cycling with VC and FEC as additives.

4) *Surface functionality- Most silicon surface has a native oxide layer which could be responsible for lower ICE. Even 80% ICE reported is not a good enough in long run although it's better than 70 or 75% reported in other studies. Surface functionality is one the key factors for SEI composition and stability. The authors should emphasize a few studies on this their work - ACS Appl. Mater. Interfaces., 2014, 6 (10) 7607 and few others. Investigation the surface chemical composition could help to understanding the ICE during the initial cycles.*

Our reply: We greatly appreciate the reviewer’s valuable suggestions. Indeed, the surface composition and morphology can affect the SEI composition and stability as well as ICE. In this revision, we cite ACS Appl. Mater. Interfaces., 2014, 6 (10) 7607-7610 as Ref. 13 and added other two papers (ACS Nano 9, 2015,6576-6586; Energy Storage Mater. 2018, 15, 139-147) on surface functionality of Si as Refs 18 and 26.

We have provided the XPS results (Supplementary Fig. 3 and Fig. R14 in this reply letter) of AMPSi and AMPSi@C. The presence of SiO_x species in both AMPSi and AMPSi@C is shown and the oxygen content is about 6.7 % (wt.). The native SiO_x layer reduces the ICEs of AMPSi and AMPSi@C. Our study shows that AMPSi has a high ICE of 86.6% and ICE of AMPSi@C is slightly reduced to 80.3% due to the low crystallinity and increased surface area of the continuous carbon coating.

Fig. R14. XPS survey and corresponding high-resolution Si 2p spectra of AMPSi (a and b) and AMPSi@C (c and d).

5) Raman- AMPSi show peaks around 509 cm^{-1} . Its neither amorphous nor crystalline Is? Crystalline or semicrystalline peak should have a peak around 521 cm^{-1} and amorphous Si should be a weak feature around 580 cm^{-1} . This needs to be looked. Even strained Si should not show such large shift compared to crystalline peak at 521 cm^{-1} .

Our reply: We thank the reviewer's valuable comments. We have repeated the Raman analysis on different AMPSi samples and all the measured Raman peaks corresponding to first-order optical phonon of AMPSi are centered at $509\text{-}511\text{ cm}^{-1}$. The Raman spectra (Fig. 3d in the revised manuscript and R15 in this reply letter) are recorded using a 532 nm laser from DXR Laser Co-Focal Microscopy Raman Spectrometer (American Thermo Electron).

Indeed, the first-order optical phonon of crystalline or semicrystalline bulk Si appears at 521 cm^{-1} . However, the Raman peak of crystalline Si is easily affected by the Si grain size and surface composition.³⁰⁻³⁴ With decreasing particle size, the frequency shifts towards smaller energies.^{35,36} Therefore, nanosized Si shows a blue-shifted Raman peak and the peak of the first-order optical phonon is generally located at $508\text{-}520\text{ cm}^{-1}$ due to phonon confinement.³⁴⁻³⁵ AMPSi is composed of 3D interconnected crystalline Si nanoligaments with a width of several ten nanometers as confirmed by HR-TEM. Therefore, it is reasonable for the sharp Raman absorption

peak to be around 510 cm^{-1} for AMPSi. Compared to AMPSi, AMPSi@C exhibits a larger blue shift at around 504 cm^{-1} , which is attributed to the carbon coating and strong bonding between the coated carbon and Si as confirmed by Fourier transform infrared (FTIR) spectroscopy (Supplementary Fig. 7 and Fig. R16 in the reply letter). The bands at 1060 and 1620 cm^{-1} correspond to the characteristic vibrations of the function groups on Si. Compared to AMPSi, the band at 1060 cm^{-1} is split into two peaks at 1240 and 1090 cm^{-1} in AMPSi@C, suggesting robust bonding at the interface between Si and coated carbon shell³. In the revision, we have added the discussion on the Raman shifts of AMPSi and AMPSi@C as highlighted in red in the revised manuscript.

Fig.R15. Raman spectra of AMPSi and AMPSi@C.

Fig. R16. FTIR spectra of the AMPSi and AMPSi@C.

Reviewer #3 (Remarks to the Author):

In this work, a new protocol has been developed to produce high-performance Si-based materials for Li-ion batteries. The 3D interconnected Si nanoligaments and nanopores of this material prevent its excessive swelling and cracking upon lithiation, resulting in excellent electrochemical performance. The strategy to produce nanostructured/nanoporous Si particles likely to buffer the Si expansion associated with its lithiation is not new. However, the proposed protocol has the advantage to be simple, scalable and yields Si particle with a tap density larger than usual Si nanopowders. The study is well conducted and the obtained electrochemical performances are among the best published to date for Si-based anodes. However, some corrections are required as detailed hereafter:

Our reply: Thank you very much for your careful and insightful review of our manuscript. The referee's encouraging suggestions have been carefully considered and thoroughly addressed as shown below.

1) *It is known that certain variability exists in the electrode capacity measurements for similar experiments (typically around $\pm 5\%$ from my own experiments). Thus, this makes no sense to present the capacity values with one digit after the decimal point, as done through the manuscript.*

Our reply: Thanks for your valuable suggestion. Indeed, there is certain variability in the electrode capacity measurements. In the revision, we have removed one digit after the decimal point from the Fig.s and throughout the manuscript.

2) *line 35: "comprising the AMPSi@C anode" must be replaced by "comprising prelithiated AMPSi@C anode"*

Our reply: We thank the reviewer for this good suggestion. We have corrected it.

3) *Line 43: the theoretical specific capacity of Si is not ($Li_{22}Si_5$) but 3579 mAh/g (i.e. $Li_{15}Si_4$) since $Li_{22}Si_5$ cannot be formed electrochemically as clearly demonstrated by Dahn et al. several years ago. (see Journal of The Electrochemical Society, 151 (2004) A838-A842).*

Our reply: We greatly appreciate the reviewer's valuable comment. $\text{Li}_{22}\text{Si}_5$ with 4200 mAh/g can only be formed at a high temperature³⁷ and the theoretical specific capacity of Si should be 3579 mAh/g ($\text{Li}_{15}\text{Si}_4$) when Si is full lithiated at room temperature. In this revision, we have revised the sentence in page 3 as “*the theoretical specific capacity of Si is 3579 mAh/g ($\text{Li}_{15}\text{Si}_4$)*” and cited the paper in Journal of The Electrochemical Society, 151 (2004) A838-A842 as Ref. 3.

4) *It would be relevant to measure the O content in the AMPSi and AMPSi@C powders to confirm that they are not oxidized during or after their synthesis.*

Our reply: We thank the reviewer for this constructive suggestion. We have conducted XPS characterization to confirm the surface chemical states of AMPSi and AMPSi@C and the results are provided as Supplementary Fig. 3 (Fig. R17 in this reply letter). The full XPS and fine Si 2p spectrum of AMPSi indicate the presence of O species. The two peaks at 98.8 and 99.4 eV are attributed to Si 2p_{3/2} and Si 2p_{1/2} of crystalline Si and those at 102.2 and 102.9 eV are ascribed to Si-O due to surface oxidation after synthesis.^{38,39} The XPS data of AMPSi@C reveal weak SiO_x and the O content is measured to be 6.7% (wt.)

Fig. R17. XPS survey and corresponding high-resolution Si 2p spectra of AMPSi (a and b) and AMPSi@C (c and d).

5) As shown in Fig. 4a, a large capacity decay is observed during the first ~20 cycles for all electrodes. The origin of this decay must be discussed. Moreover, this is in contradiction with the CV curves presented in supplementary Fig. 9 suggesting an activation of the electrodes during the first cycles. Actually, it would be more relevant to show the evolution with cycling of selected dQ/dV curves from the Fig 4a experiments.

Our reply: Thanks for your valuable suggestions. In Fig. 4a, the first 3 cycles are measured at 0.05 C and following cycles are set at 0.5 C. The large capacity decay during the first ~20 cycles is attributed to the increased current density from 0.05 to 0.5 C and continuous formation of SEI due to the slow penetration of a viscous organic electrolyte into the continuous porous structure in AMPSi@C as a result of the strong capillary effect, volume expansion of Si, and low crystalline carbon coating. Actually, the capacity decay during the first several and even tens of cycles have been observed from Si and other anodes materials.^{3,40,41} Different from deep charging/discharging galvanostatic cycling from 1 to 0.01 V, CV is conducted at a slower scanning rate of 0.1 mV/s. Considering that the lithium alloying/dealloying process brings about significant internal structural changes (disorders) in the Si electrodes, the transport rate of Li into crystalline silicon to form amorphous Si-Li alloy is postponed by reconstruction of structure, which causes the gradual activation of the AMPSi@C electrodes during the first CV cycles⁴²⁻⁴⁴. In the revision, we provide the dQ/dV curves during the first 50 cycles in Supplementary Fig. 12 (Fig. R18 in this reply letter). It is found that the anodic and cathodic peaks overlap after 20 cycles suggesting high cycle reversibility^{45,46}.

Fig. R18. Plots of differential capacity of AMPSi@C in different cycles

6) *The areal mass loading of the electrodes should be indicated in Fig4a-c captions as it has a major impact on the capacity decay as shown in Fig. 4d. The areal capacity of the full-cell presented in Fig. 4e,f and supplementary Fig. 11 must also be indicated*

Our reply: We appreciate the reviewer's suggestion. We have added the areal mass loading of the electrodes to the Figure captions of Fig. 4 and Supplementary Fig. 11 in the revised text as highlighted in red color.

7) *Line 231: the authors indicate a volume capacity of 1712 mAh/cm³ at 0.1C. It must be indicated if this value is determined at the lithiated stage (i.e by considering the volume expansion of the electrode). Information on the cycle number and the areal capacity (or areal mass loading) corresponding to this volume capacity must be also added.*

Our reply: Thank you very much for the reviewer's encouraging comment. The volume capacity of 1,712 mAh cm⁻³ at 0.1 C is determined in the lithiated stage by considering the volume expansion of the electrode. The areal mass loading is 0.8 mg cm⁻² and cycle number is 100 cycles. When the areal mass loading is increased to 2.9 mg cm⁻², the volume capacity at the lithiated state could reach 1,760 mAh cm⁻³ at 0.1 C (Supplementary Table 1). In the revision, we have added the cycle number and areal mass loading corresponding to the volume capacity of 1712 mAh/cm³ as highlighted in red in the text.

8) *Lines 238-239: "...these values are the best hitherto reported from Si anodes". This is not true. Larger areal capacities (10 mAh/cm² for +400 cycles) have been obtained by Mazouzi et al. (see Adv. Energy Mater. 2014, 1301718)*

Our reply: We are grateful to the reviewer's helpful suggestion. Mazouzi et al.⁴⁷ have reported a larger areal capacity of 10 mAh cm⁻² by using Cu foam as current collectors and carbon nanofibers as a conductive additive. The large areal capacity of 10 mAh cm⁻² is achieved at a high Si mass loading of 10 mg cm⁻² and 300 μm thick electrode. In our work, the areal capacity of 7.1 mAh cm⁻² is achieved at an areal mass loading of 2.9 mg cm⁻² and the thickness of AMPSi@C anode is 45.1 μm. In the revision, we have revised the sentence "...these values are the best hitherto reported from Si anodes" as "...these values are higher compared to most reported Si

anodes”

9) Lines 260-272. *The full-cell cycling tests are performed after prelithiation and preactivation of the Si electrodes in half-cells. This procedure is not compatible with battery manufacturing procedures, which prevents the implementation of the present Si electrodes in commercial batteries. This point MUST be discussed in the manuscript. Cycling performance of full-cells obtained with no prelithiated/preactivated Si anodes should be also shown.*

Our reply: We thank the reviewer for the constructive suggestion. Since the initial Coulombic efficiency (ICE) of AMPSi@C anode is 80.6%, we carried out prelithiation of the anode in half cells with AMPSi@C as the working electrode and Li foil as the counter electrode. The prelithiation process is only conducted *via* the 1st discharging process and the working electrode (AMPSi@C) is lithiated to 0.01 V at a 0.05 C rate by a galvanostatic discharging method and the potential is kept for 30 min. After prelithiation, the half-cell of AMPSi@C//Li is disassembled in a glove box and the AMPSi@C electrode is quickly taken out and assembled with the Li(Ni_{1/3}Co_{1/3}Mn_{1/3})O₂ (NCM) cathode to form a full cell.

In the revision, we have added the cycling performance of full cells obtained with no prelithiated AMPSi@C anodes (Supplementary Fig. 20b and Fig. R19 in this reply letter). The full cell shows an initial Coulombic efficiency (CE) of 83.1% and delivers a lower capacity and poor cycle stability compared to prelithiated AMPSi@C anodes, indicating the pre-lithiation strategies is necessary to enhance the performance of the full cell. We agree that the prelithiation procedures described here are not compatible with large-scale manufacturing of commercial batteries. However, our work provides the data to support the importance of pre-lithiation to achieve long-term cycle stability in a full cell although the AMPSi@C shows superior electrochemical properties in a half-cell with the Li foil as the counter electrode. Scalable pre-lithiation strategies and technologies (such as pre-lithiation by direct contact to lithium metal or use of lithiated active materials as negative electrode additives) have recently been reported for Si anodes⁴⁸⁻⁵¹ and they can be implemented in prelithiation of the AMPSi@C anodes for commercial batteries.

In this revision, we have provided the cycling performance of the full cell with no prelithiated AMPSi@C anode in Supplementary Fig. 20b and added the prelithiation discussion as highlighted in red in the revised text.

Fig. R19. Cycling performance of full cell with no pre-lithiated Si as anode at 0.5C and corresponding CE

References:

- 1 Wei, S., Liu, W., Li, X., Zhao, X. & Yan, X. Bean pod-like Si@dopamine-derived amorphous carbon@N-doped graphene nanosheet scrolls for high performance lithium storage. *J. Mater. Chem. A* **4**, 10948-10955 (2016).
- 2 Liu, N. *et al.* A yolk-shell design for stabilized and scalable Li-ion battery alloy anodes. *Nano Lett.* **12**, 3315-3321 (2012).
- 3 Jing, W. *et al.* Scalable synthesis of defect abundant Si nanorods for high-performance Li-Ion battery anodes. *ACS Nano* **9**, 6576-6586 (2015).
- 4 Huo, K. *et al.* Mesoporous nitrogen-doped carbon hollow spheres as high-performance anodes for lithium-ion batteries. *J. Power Sources* **324**, 233-238 (2016).
- 5 Dong, J. L. *et al.* Nitrogen-doped carbon coating for a high-performance SiO anode in lithium-ion batteries. *Electrochem. Commun.* **34**, 98-101 (2013).
- 6 Czerw, R. *et al.* Identification of electron donor states in N-doped carbon nanotubes. *Nano Lett.* **1**, 457-460 (2001).
- 7 Xiao, K. *et al.* n-Type field-effect transistors made of an individual

- nitrogen-doped multiwalled carbon nanotube. *J. Am. Chem. Soc.* **127**, 8614-8617 (2005).
- 8 Zhong, Z., Lee, G. I., Chan, B. M., And, S. H. H. & Kang, J. K. Tailored field-emission property of patterned carbon nitride nanotubes by a selective doping of substitutional N(sN) and pyridine-like N(pN) atoms. *Chem. Mater.* **19**, 2918-2920 (2007).
- 9 Seong Ho, Y., Weon Ho, S. & Jeung Ku, K. The nature of graphite- and pyridinelike nitrogen configurations in carbon nitride nanotubes: dependence on diameter and helicity. *Small* **4**, 437-441 (2010).
- 10 Black, D. L., Mcquay, M. Q. & Bonin, M. P. Laser-based techniques for particle-size measurement: A review of sizing methods and their industrial applications. *Prog. Energ. Combust.* **22**, 267-306 (1996).
- 11 Wei, L. *et al.* Surface and interface engineering of silicon-based anode materials for lithium-ion batteries. *Adv. Energy Mater.* **7**, 1701083 (2017).
- 12 Wang, B., Li, W., Tian, W., Jing, G. & Wen, Z. Self-template construction of mesoporous silicon submicrocube anode for advanced lithium ion batteries. *Energy Storage Mater.* **15**, 139-147 (2018).
- 13 Chen, S., Bao, P., Huang, X., Sun, B. & Wang, G. Hierarchical 3D mesoporous silicon@graphene nanoarchitectures for lithium ion batteries with superior performance. *Nano Res.* **7**, 85-94 (2014).
- 14 Quan, X. *et al.* Watermelon-inspired Si/C microspheres with hierarchical buffer structures for densely compacted lithium-ion battery anodes. *Adv. Energy Mater.* **7**, 1601481 (2016).
- 15 Kim, J. S. *et al.* Three-dimensional silicon/carbon core-shell electrode as an anode material for lithium-ion batteries. *J. Power Sources* **279**, 13-20 (2015).
- 16 Croce, F., Focarete, M. L., Hassoun, J., Meschini, I. & Scrosati, B. A safe, high-rate and high-energy polymer lithium-ion battery based on gelled membranes prepared by electrospinning. *Energ. Environ. Sci.* **4**, 921-927 (2011).
- 17 Jung, H. G., Hassoun, J., Park, J. B., Sun, Y. K. & Scrosati, B. An improved

- high-performance lithium–air battery. *Nat. Chem.* **4**, 579-585 (2012).
- 18 Jin, Y. *et al.* Understanding fluoroethylene carbonate and vinylene carbonate based electrolytes for Si anodes in lithium ion batteries with NMR spectroscopy. *J. Am. Chem. Soc.* **140**, 9854-9867 (2018).
- 19 Yuan, Y., Amine, K., Lu, J. & Shahbazianyassar, R. Understanding materials challenges for rechargeable ion batteries with in situ transmission electron microscopy. *Nat. Commun.* **8**, 15806 (2017).
- 20 Nie, A. *et al.* Lithiation-induced shuffling of atomic stacks. *Nano Lett.* **14**, 5301-5307 (2014).
- 21 Liu, X. H. *et al.* In situ TEM experiments of electrochemical lithiation and delithiation of individual nanostructures. *Adv. Energy Mater.* **2**, 722-741 (2012).
- 22 Qi, G. *et al.* Direct evidence of lithium-induced atomic ordering in amorphous TiO₂ nanotubes. *Chem. Mater.* **26**, 1660-1669 (2014).
- 23 Gu, M. *et al.* Demonstration of an electrochemical liquid cell for operando transmission electron microscopy observation of the lithiation/delithiation behavior of Si nanowire battery anodes. *Nano Lett.* **13**, 6106-6112 (2013).
- 24 Liu, X. H. *et al.* Ultrafast electrochemical lithiation of individual Si nanowire anodes. *Nano Lett.* **11**, 2251-2258 (2011).
- 25 Seok Woo, L., McDowell, M. T., Jang Wook, C. & Yi, C. Anomalous shape changes of silicon nanopillars by electrochemical lithiation. *Nano Lett.* **11**, 3034-3039 (2011).
- 26 Goldman, J. L., Long, B. R., Gewirth, A. A. & Nuzzo, R. G. Strain anisotropies and self-limiting capacities in single-crystalline 3D silicon microstructures: Models for high energy density lithium-ion battery anodes. *Adv. Funct. Mater.* **21**, 2412-2422 (2011).
- 27 Igor, K. *et al.* A major constituent of brown algae for use in high-capacity Li-ion batteries. *Science* **334**, 75-79 (2011).
- 28 Ni, S., Ma, J., Zhang, J., Yang, X. & Zhang, L. The electrochemical performance of commercial ferric oxide anode with natural graphite adding

- and sodium alginate binder. *Electrochim. Acta* **153**, 546-551 (2015).
- 29 Francesca Bigoni, F. D. G., Francesca Soavi, and Catia Arbizzani. Sodium alginate: A water-processable binder in high-voltage cathode formulations. *J. Electrochem. Soc.* **164**, A6171-A6177 (2017).
- 30 Vendamani, V. S. *et al.* Synthesis of ultra-small silicon nanoparticles by femtosecond laser ablation of porous silicon. *J. Mater. Sci.* **50**, 1666-1672 (2014).
- 31 Zi, J. *et al.* Raman shifts in Si nanocrystals. *Appl. Phys. Lett.* **69**, 200-202 (1996).
- 32 Dogan, I. L. & Sanden, M. C. M. V. D. Direct characterization of nanocrystal size distribution using Raman spectroscopy. *J. Appl. Phys.* **114**, 134310 (2013).
- 33 Wang, B. *et al.* High volumetric capacity silicon-based lithium battery anodes by nanoscale system engineering. *Nano Lett.* **13**, 5578-5584 (2013).
- 34 Smit, C., Swaaij, van, R. A. C. M. M., Donker, H., Petit, A. M. H. N., Kessels, W. M. M., & Sanden, van de, M. C. & Ma, J. Determining the material structure of microcrystalline silicon from Raman spectra. *J. Appl. Phys.* **94**, 3582-3588 (2003).
- 35 Ryu, J., Hong, D., Choi, S. & Park, S. Synthesis of ultrathin Si nanosheets from natural clays for lithium-ion battery anodes. *ACS Nano* **10**, 2843-2851 (2016).
- 36 Nie, P. *et al.* Graphene caging silicon particles for high-performance lithium-ion batteries. *Small* **14**, 1800635 (2018).
- 37 Huggins, R. A. Lithium alloy negative electrodes. *J. Power Sources* **81-82**, 13-19 (1999).
- 38 Gauthier, M. *et al.* A low-cost and high performance ball-milled Si-based negative electrode for high-energy Li-ion batteries. *Energ. Environ. Sci.* **6**, 2145-2155 (2013).
- 39 Philippe, B. *et al.* Nanosilicon electrodes for lithium-ion batteries: interfacial mechanisms studied by hard and soft X-ray photoelectron spectroscopy. *Chem.*

- Mater.* **24**, 1107-1115 (2017).
- 40 Li, Y. *et al.* Growth of conformal graphene cages on micrometre-sized silicon particles as stable battery anodes. *Nat. Energy* **1**, 16017 (2016).
- 41 Liu, N. *et al.* A pomegranate-inspired nanoscale design for large-volume-change lithium battery anodes. *Nat. Nanotechnol.* **9**, 187-192 (2014).
- 42 Ping, N. *et al.* Graphene caging silicon particles for high-performance lithium-ion batteries. *Small* **14**, 1800635 (2018).
- 43 Chan, C. K. *et al.* High-performance lithium battery anodes using silicon nanowires. *Nat. Nanotechnol.* **3**, 31-35 (2008).
- 44 Magasinski, A. *et al.* High-performance lithium-ion anodes using a hierarchical bottom-up approach. *Nat. Mater.* **9**, 353-358 (2010).
- 45 Shin, H. C., Corno, J. A., Gole, J. L. & Liu, M. Porous silicon negative electrodes for rechargeable lithium batteries. *J. Power Sources* **139**, 314-320 (2005).
- 46 Yoon, T., Nguyen, C. C., Seo, D. M. & Lucht, B. L. Capacity fading mechanisms of silicon nanoparticle negative. *J. The Electrochem. Soc.* **162**, A2325-A2330 (2015).
- 47 Mazouzi, D. *et al.* Very high surface capacity observed using Si negative electrodes embedded in copper foam as 3D current collectors. *Adv. Energy Mater.* **4**, 1079-1098 (2014).
- 48 Holtstiege, F., Bärman, P., Nölle, R., Winter, M. & Placke, T. Pre-lithiation strategies for rechargeable energy storage technologies: concepts, promises and challenges. *Batteries* **4**, 1-39 (2018).
- 49 Jie, Z. *et al.* Dry-air-stable lithium silicide–lithium oxide core–shell nanoparticles as high-capacity prelithiation reagents. *Nat. Commun.* **5**, 5088 (2014).
- 50 Zhao, J. *et al.* Artificial solid electrolyte interphase-protected Li_xSi nanoparticles: An efficient and stable prelithiation reagent for lithium-Ion batteries. *J. Am. Chem. Soc.* **137**, 8372-8375 (2015).

- 51 Zhao, J. *et al.* Metallurgically lithiated SiO_x anode with high capacity and ambient air compatibility. *P. Natl. Acad. Sci. USA* **113**, 7408-7413 (2016).

EVIEWERS' COMMENTS:

Reviewer #1 (Remarks to the Author):

The response to reviewers is one of the most comprehensive responses I have ever received and completely addresses all the queries in my review and in my opinion those of my co-reviewers. The revised manuscript is suitable for publication in Nature Communications without further revisions.

Reviewer #2 (Remarks to the Author):

The revised manuscript is much improved. The new data and analysis are excellent additions as per reviewer's suggestion. The 3D tomographic reconstructions showing the pore connectivity and microstructure adds significant insights apart from in-situ TEM work. The comparison between VC and FEC showing similar cycle life performance is very interesting. Overall the work is high quality and will be interesting to the community working in this topic.

Reviewer #3 (Remarks to the Author):

The responses to the reviewers' comments are appropriate and the revised version is acceptable for publication in my opinion.

REVIEWERS' COMMENTS:

Reviewer #1 (Remarks to the Author):

The response to reviewers is one of the most comprehensive responses I have ever received and completely addresses all the queries in my review and in my opinion those of my co-reviewers. The revised manuscript is suitable for publication in Nature Communications without further revisions.

Our Reply: Thanks for the reviewer's positive comments.

Reviewer #2 (Remarks to the Author):

The revised manuscript is much improved. The new data and analysis are excellent additions as per reviewer's suggestion. The 3D tomographic reconstructions showing the pore connectivity and microstructure adds significant insights apart from in-situ TEM work. The comparison between VC and FEC showing similar cycle life performance is very interesting. Overall the work is high quality and will be interesting to the community working in this topic.

Our Reply: We appreciate the reviewer's positive comments.

Reviewer #3 (Remarks to the Author):

The responses to the reviewers' comments are appropriate and the revised version is acceptable for publication in my opinion.

Our Reply: Thanks for the referee's encouraging comments